# Meta-learning biologically plausible plasticity rules with random feedback pathways

Navid Shervani-Tabar[1] ✉ & Robert Rosenbaum [1]

Backpropagation is widely used to train artificial neural networks, but its relationship to synaptic plasticity in the brain is unknown. Some biological models of backpropagation rely on feedback projections that are symmetric with feedforward connections, but experiments do not corroborate the existence of such symmetric backward connectivity. Random feedback alignment offers an alternative model in which errors are propagated backward through fixed, random backward connections. This approach successfully trains shallow models, but learns slowly and does not perform well with deeper models or online learning. In this study, we develop a meta-learning approach to discover interpretable, biologically plausible plasticity rules that improve online learning performance with fixed random feedback connections. The resulting plasticity rules show improved online training of deep models in the low data regime. Our results highlight the potential of meta-learning to discover effective, interpretable learning rules satisfying biological constraints.

Error-driven learning in multilayer neural networks was revolutionized by the error backpropagation algorithm[1], or backprop for short. In backprop, gradients or "errors" are propagated backward through auxiliary feedback pathways to compute parameter updates.

While practical, backprop has strong structural constraints that make it biologically implausible[2,3]. A major limitation, known as the weight transport problem[4] states that transmitting gradients to upstream layers requires feedback connections that are symmetric with feedforward connections. Such symmetric connectivity is not known to exist in the brain. In an attempt to depart from the symmetry assumption, Lillicrap et al.[5] show that even random backward connections can transmit effective teaching signals to train the upstream layers. In this scenario, while the backward connections are fixed, forward weights evolve to align the teaching signals with those prescribed by the backprop algorithm. However, leaving out the symmetry constraint comes with caveats. Random feedback alignment struggles with deeper networks, limited training data sizes, convolutional layers, and online data streams[6,7].

To improve random feedback alignment, Nøkland[8] proposed to rewire the feedback connections and feed the teaching signals directly from the output layer to the upstream layers. While this improves the transmission of errors, it still does not perform as robustly as the symmetric case in the low data regime. Parallel to this, Liao et al.[9] suggested dismissing symmetry in magnitude, but assigning symmetric signs to the feedback connections. Nonetheless, they found that decreasing the batch size of the training data may deteriorate performance when discarding symmetry. In addition, they found batch normalization[10] critical for training with asymmetric connections. These findings render the methods inadequate for training with an online stream of data, where the batch size is one, and ultimately undercuts their biological plausibility.

An alternative strategy is to implement a secondary update rule to modify the backward connections along with the forward weights. To that end, Akrout et al.[11] proposed to use a Hebbian plasticity rule[12] to adjust the feedback matrices parallel to the approximate gradient-based update of the forward path. The former pushes the backward connections toward the transpose of the forward weights. However, Kunin et al.[13] show that this approach is highly sensitive to hyperparameter tuning. Instead, they redefine the optimization objective as a loss function based on the forward path in combination with layer-

[1]Department of Applied and Computational Mathematics and Statistics, University of Notre Dame, Notre Dame, IN 46556, USA. ✉e-mail: nshervan@nd.edu

wise regularization terms for backward weights to update forward and backward pathways concurrently. They propose a few regularization terms and show that combining these units can achieve more stable plasticity rules.

Meta-learning is a broad learning framework consisting of a learning process that envelopes another optimization loop and learns some aspect of the inner learning procedure, effectively "learning to learn." Although this concept has been around for decades[14], Finn et al.[15] popularized meta-learning for few-shot learning applications. This approach employs meta-learning to optimize an internal representation of the network, which is subsequently used as an initial weight to expedite learning on a downstream task. Further, Javed and White[16] extended this approach to continual learning by modifying the objective function of the outer optimization loop. Still, they used the modified approach to learn a partial initialization of the network's forward weights. Although effective in learning representations for the few-shot learning, they effectively work by pre-training a model rather than by learning to learn. More precisely, their effectiveness is largely derived from their ability to meta-learn a weight initialization, rather than meta-learning a learning rule itself.

The meta-learning framework has provided a new direction for building biologically plausible computational neural models. For example, Lindsey et al.[17] learn the direct feedback pathways that modulate activations and use a supervised adaptation of Oja's rule to update forward connections. It is supervised because it benefits from modulated activations, which do not guarantee the established properties of conventional Oja's rule[18]. Nevertheless, they also meta-learn an initial value for the forward connections, which makes their approach dependent on the learned weight initialization, not on the learned learning rule alone. Miconi et al.[19,20] showed that meta-learning can train a variety of network architectures on various tasks. Like Lindsey et al.[17], their approach meta-trains a separate plasticity rule for each weight. While this approach can be effective, the resulting plasticity rules are difficult to interpret. In addition, meta-learning weight initialization in these works makes it unclear to what degree the results are affected by the proposed plasticity rule as opposed to the weight initialization.

A growing body of work aims to only meta-learn a plasticity rule without inferring any component of the inner model, such as initial weights. Early work includes Bengio et al.[21], who meta-learned a parametric learning rule to train a 2D classifier and boolean function. In each meta-iteration, they used the plasticity rule to train multiple networks on separate tasks and obtained the meta-loss function by summing over the loss of all these networks. More recent work includes Andrychowicz et al.[22], who parametrize the learning rule with a Recurrent Neural Network (RNN) and meta-learn weights of the RNN model. Using an RNN allows for training a dynamic update rule. In the context of biological plausibility, Confavreux et al.[23] used meta-learning to determine plasticity rules that train shallow linear networks. Rather than discovering new rules, they recover well-known plasticity rules using objective functions based on their known behavior.

The scope of the meta-learning framework is beyond learning the forward pathway's plasticity rule. Meta-learning has given rise to unorthodox training models beyond the classic backward transmission of errors. For example, Metz et al.[24] used a meta-learning framework to learn a plasticity rule for unsupervised learning. They proposed to infer the teaching signals by meta-learning a network that projects forward activation units and the downstream feedback signal into backward hidden states. These hidden states are subsequently used to update the forward and backward weights via each pathway's meta-learned plasticity rule. Another related work on semi-supervised learning[25] uses learnable auxiliary feedback and lateral connections to facilitate error propagation during training and meta-learns the plasticity rules to update these connections. Finally, Sandler et al.[26]

reformulate the interactions between the forward and backward activations by defining parameterized update rules for both feedforward and feedback connections. Then, they yield new plasticity rules by meta-learning these hyperparameters.

Here, we improve upon previous work by discovering a plasticity rule that enhances the flow of information in the backward pathway while learning more distinctive embeddings in the forward network. We use meta-learning to learn a parameterized plasticity rule based on a combination of candidate rules. Key features that characterize our approach include:

1. Our approach solely meta-learns a plasticity rule and does not learn a weight initialization. As a result, our approach learns a learning rule that can be applied to train "naive," randomly initialized networks from scratch.
2. We use "meta-parameter sharing" in the sense that all weights share a common plasticity rule, instead of learning a separate plasticity rule for each weight. This approach allows us to interpret and understand the meta-learned plasticity rules.
3. We impose an L1 penalty on the plasticity coefficients in our meta-loss function. This encourages our algorithm to learn a plasticity rule with fewer terms, further simplifying the analysis and interpretability of the resulting rule.
4. Our inner learning loop uses online learning (batch size 1) and limited training data (250 data points). Coupled with the random weight initialization in our inner learning loop, this forces the plasticity rules to learn in a more biologically relevant and challenging setting, with which random feedback alignment is known to struggle[9].

Previous studies have employed different combinations of elements, such as meta-parameter sharing (as used in ref. [23]) and online learning (as used in ref. [17]). In contrast to previous studies, we integrate all these features to address the weight alignment problem. Our analysis of the meta-learned plasticity rules demonstrates how they overcome the weight alignment challenge. Our approach further advances the use of meta-plasticity to understand how effective learning can emerge in biological neural circuits.

## Results
### Limitations of feedback alignment in deep networks
Consider a fully connected deep neural network $f_{\boldsymbol{W}}$ parameterized by weights $\boldsymbol{W}$, representing a non-linear mapping $f_{\boldsymbol{W}}: \mathbf{x} \mapsto \mathbf{y}_L$ from the network's input $\mathbf{y}_0 = \mathbf{x}$ to the output $\mathbf{y}_L$, with $L$ denoting the depth of the network. Each network layer is defined by

$$\mathbf{z}_\ell = \boldsymbol{W}_{\ell-1,\ell}\mathbf{y}_{\ell-1}, \tag{1}$$

$$\mathbf{y}_\ell = \sigma(\mathbf{z}_\ell), \tag{2}$$

where $\mathbf{y}_\ell$ is the activation for layer $\ell$ and $\sigma$ stands for the non-linear activation function.

Given a dataset $\mathcal{D}_{\text{train}} = (\boldsymbol{X}_{\text{train}}, \boldsymbol{Y}_{\text{train}})$, the model is trained in an attempt to find the set of weight parameters $\boldsymbol{W} = \{\boldsymbol{W}_{\ell-1,\ell}|0 < \ell \leq L\}$, that minimize a loss function $\mathcal{L}(\mathbf{y}_L, \boldsymbol{Y}_{\text{train}})$. Each weight matrix $\boldsymbol{W}_{\ell-1,\ell}$ is modulated by a teaching signal $\mathbf{e}_\ell$ derived from $\mathcal{L}$. A commonly used method to compute $\mathbf{e}_\ell$ is to analytically calculate the modulatory signal $\mathbf{e}_L$ in the output layer and then use a backward auxiliary network to transmit it to the upstream layers. This backward projection follows the relation

$$\mathbf{e}_\ell = \boldsymbol{B}_{\ell+1,\ell}\mathbf{e}_{\ell+1} \odot \sigma'(\mathbf{z}_\ell), \tag{3}$$

where $\odot$ denotes element-wise multiplication and $\boldsymbol{B} = \{\boldsymbol{B}_{\ell+1,\ell}|0 < \ell < L\}$ are the set of feedback connections.

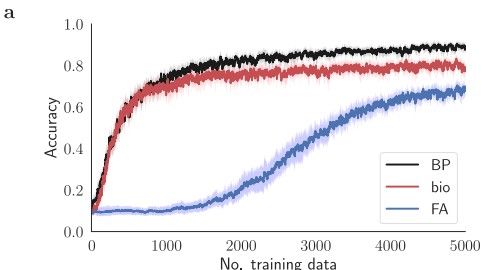
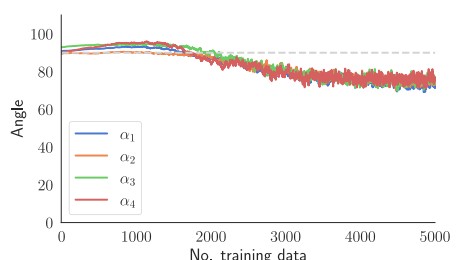

**Fig. 1 | Feedback alignment learns poorly in deep models.** Performance of benchmark learning schemes while training a 5-layer fully connected classifier network on MNIST digits[47] with online learning. **a** Accuracy versus the number of training data for Feedback Alignment (FA)[5] and backprop (BP)[1] methods, compared to the discovered biologically plausible plasticity rule (bio) presented below. **b** The angle $\alpha_\ell$ between the teaching signal $\mathbf{e}_\ell^{\text{FA}}$ transmitted by the Feedback Alignment

method and the corresponding backpropagated signal $\mathbf{e}_\ell^{\text{BP}}$ (in degrees) for different layers $\ell = 1, 2, 3$, and 4. For both approaches, $\mathbf{e}_5$ is computed using $\partial\mathcal{L}/\partial\mathbf{z}_L$ and has the same value, resulting in $\alpha_5 = 0$. In all figures, each plot illustrates the mean over multiple trials. The shaded area represents the 98% confidence interval (see Methods).

In a gradient-based optimization algorithm, $\mathbf{e}_L$ is defined as the derivative of the loss function $\mathcal{L}$ with respect to $\mathbf{z}_L$. This teaching signal is propagated backward up to the initial layer to modulate the weight parameters. A widely used scheme, backprop, uses feedback weights $\mathbf{B}_{\ell+1,\ell}^{\text{BP}}$ that are the transposes of the forward path's weights, to transport these modulating signals using Eq. (3). Subsequently, the forward weight parameters are updated by

$$\Delta\mathbf{W}_{\ell-1,\ell} = -\theta\mathbf{e}_\ell\mathbf{y}_{\ell-1}^T, \tag{4}$$

which represents a shared plasticity rule for all forward connections $\mathbf{W}_{\ell-1,\ell}$ and $\theta$ is the associated learning rate.

To alleviate the biologically undesirable characteristics of the backprop algorithm, ref. [5] proposed the "Random Feedback Alignment" approach, which departs from the assumption of symmetric feedback connections and instead uses fixed random backward connections $\mathbf{B}^{\text{FA}}$ that are not bound to the forward weights. To distinguish between the two learning algorithms, we hereafter use the phrase "feedback alignment" to refer to the learning rule in Eq. (4) with fixed random $\mathbf{B}_{\ell+1,\ell}^{\text{FA}}$ and we use "backprop" to refer to Eq. (4) with $\mathbf{B}_{\ell+1,\ell}^{\text{BP}} = \mathbf{W}_{\ell,\ell+1}^T$.

For feedback alignment, the teaching signal $\mathbf{e}_\ell^{\text{FA}}$ is not an exact gradient, but an approximating pseudo-gradient term. The resulting learning algorithm performs well on simple tasks and shallower networks. However, feedback alignment fails to reach good accuracy in deeper networks and is not as robust in the small data regime. In our empirical test with an online stream of data, feedback alignment only begins to effectively learn after about 2000 iterations, while backprop learns much more quickly (Fig. 1a). An alternative approach to using feedback connections that link consecutive layers is to create direct backward pathways[8]. This change allows errors to be transmitted directly from the output layer to the upstream layers. This modification leads to improved performance compared to the feedback alignment method, speeding up the learning process and improving accuracy. However, it still falls short of the performance level of backpropagation (see Supplementary Fig. S1). In addition, Fig. 1b shows that the teaching signals transmitted through fixed feedback connections $\mathbf{e}_\ell^{\text{FA}}$ are not aligned with the true gradients, $\mathbf{e}_\ell^{\text{BP}}$, computed by backpropagation at this stage of training.

These limitations indicate that the backward flow of information through fixed feedback is insufficient for online training in deeper models. This paper investigates modified plasticity rules to improve the trained model's performance. To that end, a meta-learning framework is adapted to explore a parameterized space of the plasticity rules.

## Meta-learning to discover interpretable plasticity rules

Meta-learning is a machine learning paradigm that aims to learn elements of a learning procedure. This framework consists of a two-level learning scheme: An inner adaptation loop that learns parameters $\mathbf{W}$ of a model $f_{\mathbf{W}}$ using a parameterized plasticity rule $\mathcal{F}(\mathbf{\Theta})$ and an outer meta-optimization loop that modifies the plasticity meta-parameters $\mathbf{\Theta}$. The meta-training dataset contains a set of tasks $\{\mathcal{T}_\varepsilon\}_{0\leq\varepsilon\leq\mathcal{E}}$, each consisting of $K$ training data $(\mathbf{X}_{\text{train}}, \mathbf{Y}_{\text{train}})$ and $Q$ query data $(\mathbf{X}_{\text{query}}, \mathbf{Y}_{\text{query}})$ per class. The former is used to train the model $f_{\mathbf{W}}$ while the latter optimizes the meta-parameters $\mathbf{\Theta}$. Algorithm 1 details the meta-learning framework presented in this work.

**Algorithm 1. Meta-learning algorithm.**
Input    meta-training    set    $\{\mathcal{T}_\varepsilon\}_{0\leq\varepsilon\leq\mathcal{E}} = \{(\mathbf{X}_{\text{train}}^\varepsilon, \mathbf{Y}_{\text{train}}^\varepsilon), (\mathbf{X}_{\text{query}}^\varepsilon, \mathbf{Y}_{\text{query}}^\varepsilon)\}_{0\leq\varepsilon\leq\mathcal{E}}$, plasticity rule $\mathcal{F}$, number of episodes $\mathcal{E}$, meta-learning rate $\eta$, and regularization coefficient $\lambda$.
Initialize learning parameters $\mathbf{\Theta}^{(0)}$.
**for** $\varepsilon = 0, \ldots, \mathcal{E}$ **do**
    Initialize network parameters $\mathbf{W}^{(0)}$ and $\mathbf{B}$.
    **for** $(\mathbf{x}_{\text{train}}^{(i)}, y_{\text{train}}^{(i)}) \in (\mathbf{X}_{\text{train}}^\varepsilon, \mathbf{Y}_{\text{train}}^\varepsilon)$ **do**
        Set $\mathbf{y}_0 = \mathbf{x}_{\text{train}}^{(i)}$
        **for** $\ell = 1, \ldots, L$ **do**
            Compute $\mathbf{z}_\ell$ (Eq. (1)).
            Compute $\mathbf{y}_\ell$ (Eq. (2)).
        **end for**
    Compute $\mathcal{L}(\mathbf{y}_L, y_{\text{train}}^{(i)})$.
    Compute $\mathbf{e}_L = \partial\mathcal{L}/\partial\mathbf{z}_L$.
    **for** $\ell = L, \ldots, 1$ **do**
        Compute $\mathbf{e}_{\ell-1} = \mathbf{B}_{\ell,\ell-1}\mathbf{e}_\ell \odot \sigma'(\mathbf{z}_{\ell-1})$ (Eq. (3)).
        Update $\mathbf{W}_{\ell-1,\ell}^{(i+1)} = \mathbf{W}_{\ell-1,\ell}^{(i)} + \mathcal{F}(\mathbf{e}_{\ell-1}, \mathbf{y}_{\ell-1}, \mathbf{e}_\ell, \mathbf{y}_\ell, \mathbf{W}_{\ell-1,\ell}^{(i)}; \mathbf{\Theta}^{(\varepsilon)})$.
    **end for**
    **end for**
    Update meta-parameters $\mathbf{\Theta}^{(\varepsilon+1)} = \mathbf{\Theta}^{(\varepsilon)} - \eta\nabla_{\mathbf{\Theta}^{(\varepsilon)}}\left[\mathcal{L}(f(\mathbf{X}_{\text{query}}^\varepsilon; \mathbf{W}^{(i+1)}), \mathbf{Y}_{\text{query}}^\varepsilon) + \lambda \| \mathbf{\Theta}^{(\varepsilon)}\|_1\right]$.
**end for**

In each meta-iteration, also known as an episode, a randomly initialized model $f_{\mathbf{W}}$ is trained on an online training data sequence. In other words, each adaptation iteration uses a single data point $(\mathbf{x}_{\text{train}}, y_{\text{train}})$ to update $\mathbf{W}$. It is worth emphasizing that reinitializing weights $\mathbf{W}$ at each episode removes the learning rule's dependence on the weight initialization. The meta-learned plasticity rules are therefore optimized to learn a task starting from a randomly initialized weight matrix. In contrast, meta-optimizing initial weights will adapt meta-parameters $\mathbf{\Theta}$ to the later stages of learning, which does not extrapolate to the training lifetime anymore. Moreover, when meta-learning a weight initialization in conjunction with a plasticity rule (e.g.[17]), it is

not clear to what extent improvements in learning can be attributed to the weight initialization versus the meta-learned plasticity rule itself.

Each episode $\varepsilon$ follows two objectives. The first is to quantify the model parameters $\boldsymbol{W}$ using a loss function $\mathcal{L}$, iteratively, on each data point sampled from task $\mathcal{T}_\varepsilon$'s training set. Then, given a set of $R$ candidate terms $\{\mathcal{F}^r\}_{0 \leq r \leq R-1}$, a parameterized plasticity rule is defined as a linear combination of individual plasticity terms,

$$\mathcal{F}(\boldsymbol{\Theta}) = \sum_{r=0}^{R-1} \theta_r \mathcal{F}^r. \tag{5}$$

where $\boldsymbol{\Theta} = \{\theta_r | 0 \leq r \leq R-1\}$ is the set of learning parameters shared across layers. This rule is used to update forward weights, $\boldsymbol{W}$, in the network. The second objective, dubbed meta-loss, assesses the meta-parameters $\boldsymbol{\Theta}$ by evaluating the loss function $\mathcal{L}$ on the query set of the same task $\mathcal{T}_\varepsilon$ using the updated model $f_{\boldsymbol{W}}$. While meta-learning over the pool of plasticity terms $\mathcal{F}(\boldsymbol{\Theta})$ yields an optimized set of meta-parameters, $\boldsymbol{\Theta}$, the resulting plasticity rule consists of too many terms which are difficult to interpret and understand and whose underlying mechanisms may overlap. Therefore, following Occam's razor, we introduce an L1 penalty on plasticity coefficients to select for a sparser set of plasticity terms. Mathematically put, the meta-loss is defined as

$$\mathcal{L}_{\text{meta}}(\boldsymbol{\Theta}) = \mathcal{L}(f_{\boldsymbol{W}}(\boldsymbol{X}_{\text{query}}), \boldsymbol{Y}_{\text{query}}) + \lambda \| \boldsymbol{\Theta} \|_1, \tag{6}$$

where $f_{\boldsymbol{W}}$ is the model updated in the adaptation loop and $\lambda$ is a predefined hyperparameter. The regularization term in Eq. (6) is the L1 norm of the meta-parameters, leading the algorithm to favor simplicity in the plasticity model (see Supplementary Fig. S5, and Table S1 for a comparison with alternative regularization approaches). While weights $\boldsymbol{W}$ are optimized using $\mathcal{F}(\boldsymbol{\Theta})$, meta-parameters $\boldsymbol{\Theta}$ are updated by a gradient-based approach. Figure 2 summarizes the problem's configuration.

## Benchmarking backprop and feedback alignment via meta-learning

Before introducing new plasticity rules, it is necessary to establish the baseline performance for the current learning models for the learning task considered here. To this end, we use the meta-learning framework to optimize the learning rate, $\theta$, in Eq. (4) for backprop and feedback alignment. Since, in these examples, the meta-learning model seeks to optimize the meta-parameter rather than selecting one term over the other, the regularization coefficient $\lambda$ in Eq. (6) is set to zero.

Figure 3a–c compares the performance of the two plasticity rules over 600 episodes. First, the reinitialized models $f_{\boldsymbol{W}}$ are trained at each episode using an online stream of $M \times K = 250$ data points. Then, the meta-accuracy and meta-loss are evaluated with the query data. Tracing the evolution of the plasticity coefficients in Fig. 3c shows that the meta-learning model converges after ~100 episodes. After convergence, the model trained with feedback alignment is, on average, about 25% accurate in its predictions, whereas the model back-propagated via symmetric feedbacks reaches an approximate accuracy of about 70% (Fig. 3a). In addition, the backpropagated model reaches considerably lower loss values as shown in Fig. 3b. The comparison shows that the former is not adequately trained with an online data stream in the small data regime. This outcome is further supported by Fig. 3d, which illustrates the poor alignment of the modulating signals in feedback alignment with the backprop analogs.

## Biologically plausible plasticity rules

The analysis in the previous section indicated a substantial performance gap between the backprop model and the pseudo-gradient rule with random feedback pathways early in the learning process. However, with the interrupted backward flow as the only distinction

between the two rules, the error in the last layer and activations still maintain proper information. Intuitively, introducing new local combinations of these terms to the plasticity rule may restore information flow and improve performance. To that end, we define a set of candidate plasticity terms and use meta-learning to uncover combinations that enhance learning. Meta-learning helps in two ways: finding the optimized set of meta-parameters for the linear combination of candidate terms and selecting the dominant plasticity terms. While the former avoids cumbersome hand-tuning of the coefficients, the latter provides a tool for systematically studying the space of learning rules.

We began by examining a set of $R = 10$ plasticity terms and combined them according to Eq. (5) to form the learning rule $\mathcal{F}^{\text{pool}}$ (see Methods and below for definitions of these rules). Figure 4a–c illustrates the performance of the model. We set the initial values of the meta-parameters $\{\theta_r\}_{1 \leq r < R}$ to 0. As seen in Fig. 4a, the model's accuracy initially resembles that of the FA model, but as the meta-optimization continues, the accuracy improves, starting around 10 episodes. By about 300 meta-iterations, the accuracy approaches that of the BP model. This trend is also echoed in Fig. 4b, where the loss initially follows that of the FA learning model but then declines and eventually becomes similar to that of the BP method. In Fig. 4c, it is demonstrated that the alignment angles of the teaching signals with their BP counterparts are improved compared to the FA model, seen in Fig. 3d. Figure 4d shows that the coefficients for all but 3 terms converge toward zero after about 600 episodes. Those three terms are a pseudo-gradient rule ($\mathcal{F}^0$), a Hebbian-like plasticity rule ($\mathcal{F}^2$), and Oja's rule ($\mathcal{F}^9$). Selecting these three terms and omitting the others gives a simpler plasticity rule of the form

$$\mathcal{F}^{\text{bio}}(\boldsymbol{\Theta}) = -\theta_0 \mathbf{e}_\ell \mathbf{y}_{\ell-1}^T - \theta_2 \mathbf{e}_\ell \mathbf{e}_{\ell-1}^T + \theta_9 (\mathbf{y}_\ell \mathbf{y}_{\ell-1}^T - (\mathbf{y}_\ell \mathbf{y}_\ell^T) \boldsymbol{W}_{\ell-1,\ell}), \tag{7}$$

where $\boldsymbol{\Theta} = \{\theta_0, \theta_2, \theta_9\}$ is the set of plasticity meta-parameters. $\mathcal{F}^{\text{bio}}$ performs similar to the $\mathcal{F}^{\text{pool}}$ (see Supplementary Fig. S2) and significantly improves the performance of the feedback alignment method in the low data regime (Fig. 1).

While the meta-learning successfully discovers $\mathcal{F}^{\text{bio}}$, it is important to interpret the plasticity rule and understand how it leads to improved learning. $\mathcal{F}^{\text{bio}}$ consists of three components: a pseudo-gradient term, a Hebbian-style error-based term, and Oja's rule. In what follows, we study the latter terms separately with the pseudo-gradient term to unveil the underlying reason behind their performance.

## Hebbian-style error-based plasticity rule

Motivated to understand the Hebbian-style error-based learning term in Eq. (7), we rerun the model using a plasticity rule that only includes the modified Hebbian term and the pseudo-gradient term, but omits the third term

$$\mathcal{F}^{\text{eHebb}}(\boldsymbol{\Theta}) = -\theta_0 \mathbf{e}_\ell \mathbf{y}_{\ell-1}^T - \theta_2 \mathbf{e}_\ell \mathbf{e}_{\ell-1}^T. \tag{8}$$

In Fig. 5, the meta-learning algorithm is used to optimize the coefficients $\theta_0$ and $\theta_2$, which are initialized to $10^{-3}$ and zero, respectively. Comparing the accuracy and the loss plot to $\mathcal{F}^{\text{bio}}$'s performance (Supplementary Fig. S2) shows that while $\mathcal{F}^{\text{eHebb}}$ demonstrates a significant improvement over $\mathcal{F}^0$ via feedback alignment, it is yet to reach that of $\mathcal{F}^{\text{bio}}$. Despite this, the teaching signals of $\mathcal{F}^{\text{eHebb}}$ are better aligned with the backprop direction than $\mathcal{F}^{\text{bio}}$'s (Supplementary Fig. S2), which indicates that the Hebbian error term is the driving force behind aligning the teaching signals in $\mathcal{F}^{\text{bio}}$.

Figure 6 illustrates how $\mathcal{F}^{\text{eHebb}}$ alters the communications between the backward and forward pathways. The diagram in Fig. 6a shows a model solely trained with the $\mathcal{F}^0$ via feedback alignment. In this scenario, the information from $\boldsymbol{B}_{2,1}$ flows to $\boldsymbol{W}_{0,1}$ through Eq. (3),

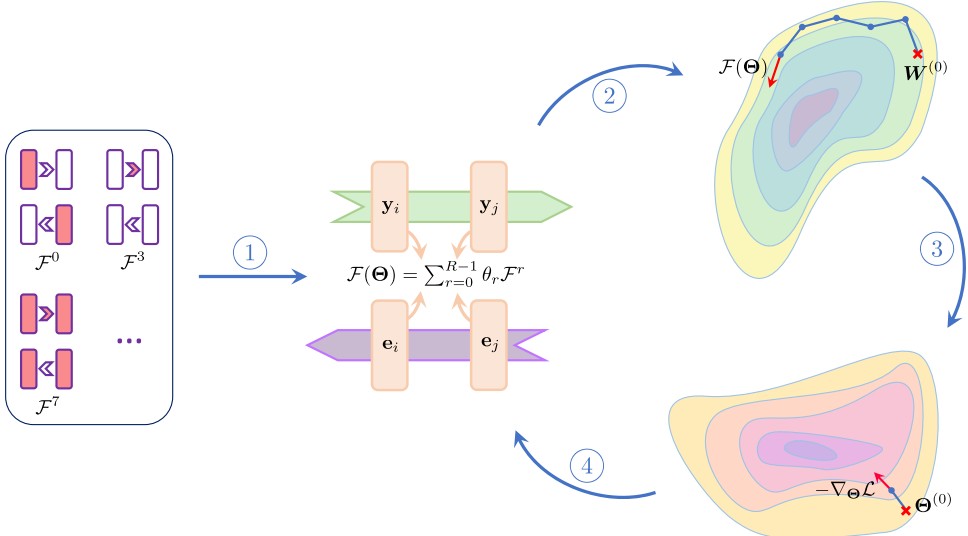

**Fig. 2 | Schematic depiction of the meta-learning workflow.** (1) A pool of $R$ biologically plausible plasticity terms $\{\mathcal{F}^r\}_{0 \leq r \leq R-1}$ is exploited to define a plasticity rule $\mathcal{F}(\boldsymbol{\Theta})$ that governs the weight updates of the model $f_{\boldsymbol{W}}$. Each term $\mathcal{F}^r$ integrates local elements available to the weight, including pre-synaptic activation $\mathbf{y}_i$, post-synaptic activation $\mathbf{y}_j$, pre-synaptic error $\mathbf{e}_i$, post-synaptic error $\mathbf{e}_j$, and the current state of the weight $W_{i,j}$. Such terms are consistent with local plasticity if $\mathbf{y}_i$ and $\mathbf{e}_i$ are encoded by the same neuron (see Discussion). The linear combination of these terms defines plasticity rule $\mathcal{F}(\boldsymbol{\Theta})$, where $\boldsymbol{\Theta} = \{\theta_r | 0 \leq r \leq R-1\}$ is the set of meta-parameters shared across the network. (2) The parameterized local learning rule $\mathcal{F}(\boldsymbol{\Theta})$ is used to navigate the weight parameter space. At each episode $\varepsilon$, $\mathcal{F}(\boldsymbol{\Theta}^{(\varepsilon)})$ iteratively searches

for optimized $\boldsymbol{W}$ starting from a random weight $\boldsymbol{W}^{(0)}$. A single data point sampled from $\mathcal{T}_\varepsilon$'s train set is used at each adaptation step for online training of the model. (3) In the meta-optimization phase, the solution $\boldsymbol{W}$ of the inner loop is used to compute the meta-loss $\mathcal{L}$ on the query set of task $\mathcal{T}_\varepsilon$. Then, a gradient-based strategy explores the meta-parameter space to optimize the plasticity meta-parameters $\boldsymbol{\Theta}^{(\varepsilon)}$. (4) The plasticity rule $\mathcal{F}(\boldsymbol{\Theta})$ is reconstructed using the updated meta-parameters $\boldsymbol{\Theta}^{(\varepsilon+1)}$ to guide the weight optimization in the next episode. This procedure is repeated until the meta-parameters converge. In the initial episodes, the unoptimized $\mathcal{F}(\boldsymbol{\Theta})$ is unlikely to direct $\boldsymbol{W}$ to a solution. However, as $\boldsymbol{\Theta}$ converges, $\mathcal{F}(\boldsymbol{\Theta})$ discovers a new direction that may only partially adhere to the direction of the gradient.

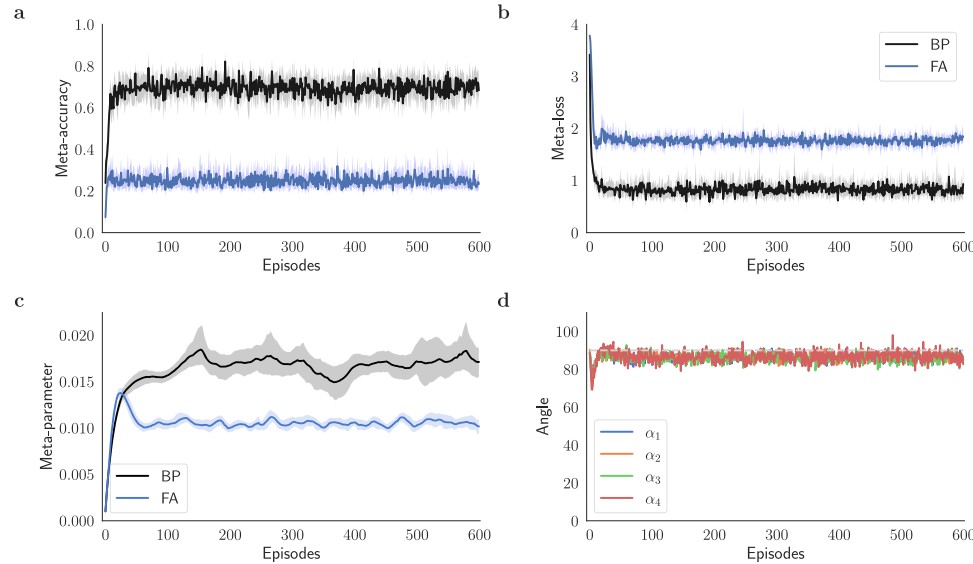

**Fig. 3 | Meta-learning plasticity coefficients for feedback alignment and backprop. a** Meta-accuracy of feedback alignment (FA) compared to backprop (BP) trained using the meta-learning framework (Alg. 1) during 600 meta-optimization episodes and **b** the corresponding meta-loss. The lower accuracy of FA compared to BP in each episode manifests slower learning of FA when presented with the same number of training examples and steps as BP for a 5-way classification task.

**c** Evolution of the learning rate meta-parameter (initialized to $10^{-3}$) with feedback alignment (FA) compared to backprop (BP) during 600 meta-optimization episodes. In this figure, each meta-parameter was optimized separately in a single-parameter meta-optimization problem and is superimposed for comparison. **d** Alignment angle $\alpha_\ell$ between modulating signals of the feedback alignment $\mathbf{e}_\ell^{\text{FA}}$ and backprop $\mathbf{e}_\ell^{\text{BP}}$ for $\ell = 1, 2, 3$, and 4. Note that $\alpha_5 = 0$ (as discussed in Fig. 1).

which is then propagated to $W_{1,2}$ after the forward pass. This configuration updates $W_{1,2}$ to align the modulator vector $\mathbf{e}_1$ with the backprop counterpart. Nonetheless, this machinery does not sufficiently align the modulating signals when applied to deeper networks with fewer training iterations. In the diagram on the right, the last layer is updated with an additional Hebbian-style plasticity term $\mathcal{F}^2$, while the

first layer is trained with vanilla $\mathcal{F}^0$ rule via feedback alignment. Once again, information from $\boldsymbol{B}_{2,1}$ flows into $W_{0,1}$. However, this time, $\mathcal{F}^{\text{eHebb}}$ introduces an auxiliary channel to flow the information from $\boldsymbol{B}_{2,1}$ to $W_{1,2}$. Finally, the forward propagation through the network implicitly transmits the information from $\boldsymbol{B}_{2,1}$ to $W_{1,2}$. The modified rule $\mathcal{F}^{\text{eHebb}}$ establishes an explicit supplementary means to communicate between

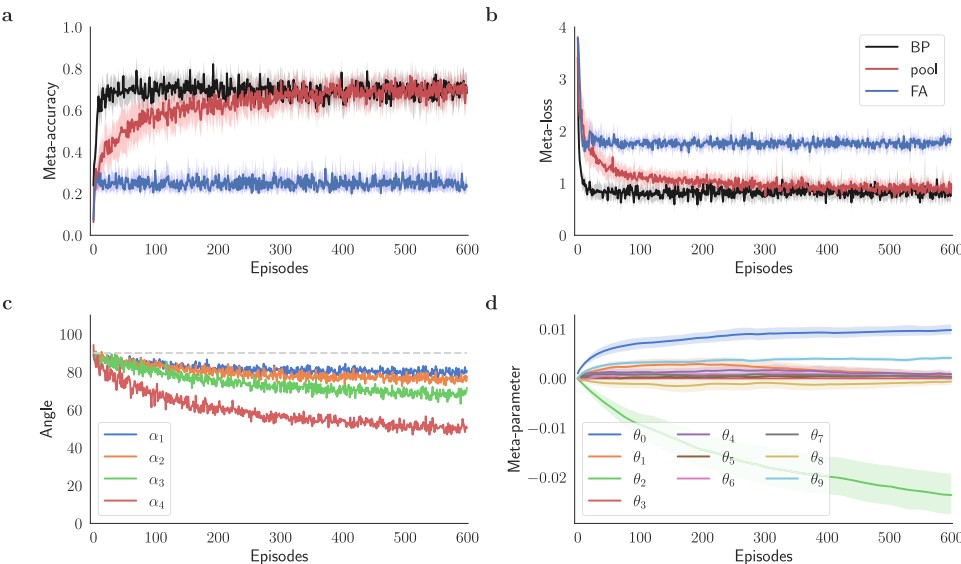

**Fig. 4 | Meta-learning biologically plausible plasticity rules.** Performance of a classifier network trained with a pool of biologically plausible plasticity rules $\mathcal{F}^{\text{pool}}$ (pool) through fixed feedback pathways. **a** Meta-accuracy and **b** meta-loss for $\mathcal{F}^{\text{pool}}$ compared to $\mathcal{F}^{0}$ via feedback alignment (FA) and backprop (BP). While learning by $\mathcal{F}^{\text{pool}}$ initially resembles $\mathcal{F}^{0}$, continued meta-optimization raises accuracy. This increase reflects the discovery of plasticity terms that can improve learning with random feedback pathways to level with the backprop method in the given

classification task. **c** Alignment angle $\alpha_\ell$ of the teaching signals of $\mathcal{F}^{\text{pool}}$ with the ones for backprop for $\ell = 1, 2, 3,$ and 4. As discussed in Fig. 1, $\alpha_5 = 0$. **d** Convergence of the plasticity coefficients $\boldsymbol{\Theta} = \{\theta_r | 0 \leq r \leq R - 1\}$ with $R = 10$. Using L1 regularization in meta-loss (Eq. (6)) sparsifies the set of meta-parameters and helps with identifying the most influential plasticity terms in learning (see Supplementary Note 6 for a discussion on alternatives).

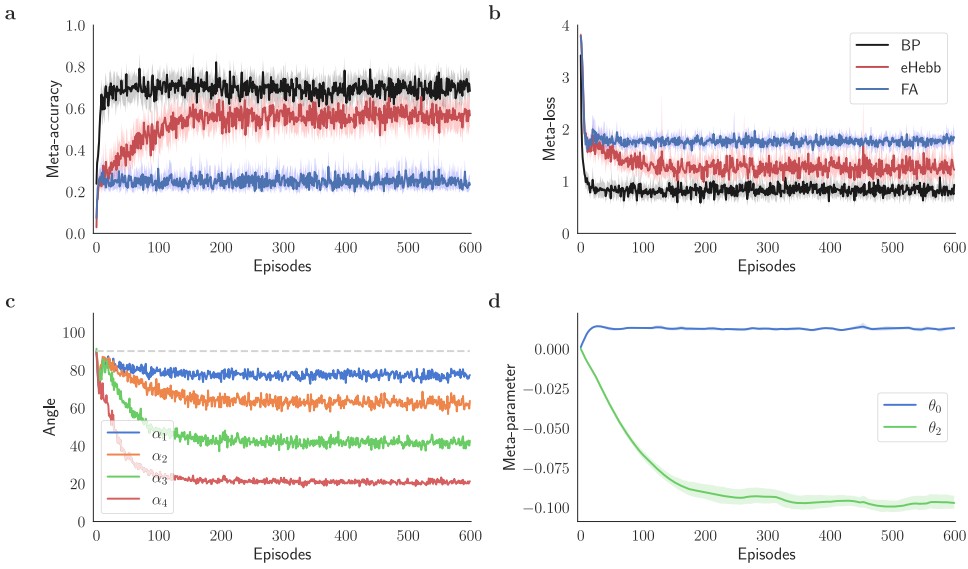

**Fig. 5 | $\mathcal{F}^{\text{eHebb}}$ improves learning through fixed feedback pathways.** Meta-training an image classifier network with $\mathcal{F}^{\text{eHebb}}$ plasticity rule (eHebb; Eq. (8)) on 5-way classification tasks sampled from EMNIST dataset[49]. **a** Meta-accuracy and **b** meta-loss plots for $\mathcal{F}^{\text{eHebb}}$ compared to $\mathcal{F}^{0}$ via feedback alignment (FA) and backprop (BP), **c** alignment angles $\alpha_\ell$ for modulating signals across the network

($\ell = 1, 2, 3,$ and 4; see Fig. 1 for $\ell = 0$) compared with backprop model. Comparing panels **a** and **c** indicates that $\mathcal{F}^{\text{eHebb}}$ improves the model's performance by rendering the modulatory signals to be more backprop-like. **d** Convergence of the plasticity coefficients $\boldsymbol{\Theta} = \{\theta_0, \theta_2\}$ using the meta-learning model (Alg. 1). The meta-optimizer starts converging after 200 episodes.

$\boldsymbol{B}_{2,1}$ and $\boldsymbol{W}_{1,2}$, boosts the alignment of $\mathbf{e}_1$, and improves the model's performance. Note that the mechanism in $\mathcal{F}^{0}$ needs two learning iterations to transmit information from $\boldsymbol{B}_{2,1}$ to $\boldsymbol{W}_{1,2}$; information from $\boldsymbol{W}_{0,1}$ propagates to $\boldsymbol{W}_{1,2}$ only after $\mathbf{y}_1$ is computed with the updated $\boldsymbol{W}_{0,1}$. Meanwhile, $\mathcal{F}^{2}$ does this in the same iteration, carrying out expedited learning.

To corroborate the argument above, we consider a 3-layer network trained with $\mathcal{F}^{0}$ rule via feedback alignment and inspect the

effect of adding the error-based Hebbian-style plasticity term $\mathcal{F}^{2}$ on the alignment angles in different layers. To that end, rather than sharing the same learning rule across the network, each layer is updated using one of the $\mathcal{F}^{0}$ rule via feedback alignment or $\mathcal{F}^{\text{eHebb}}$ rules. Table 1 determines that adding the Hebbian error term to the weight update reduces the alignment angle $\alpha$ between the pre-synaptic error and its backprop analog. A more detailed discussion can be found in Supplementary Note 3.

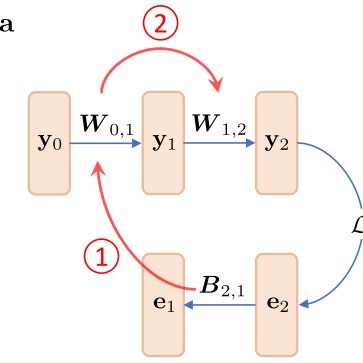

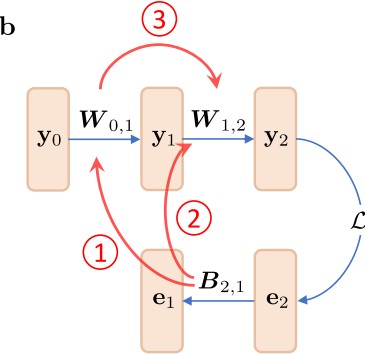

**Fig. 6 | Information flow between the forward and backward pathways. a** Both layers are trained with the rule $\mathcal{F}(\boldsymbol{\Theta}) = \theta_0 \mathcal{F}^0$ via feedback alignment. In this case, information from $\boldsymbol{B}_{2,1}$ is transmitted to $\boldsymbol{W}_{0,1}$ through $\mathcal{F}^0$ (①) and then propagated forward to $\boldsymbol{W}_{1,2}$ (②). **b** The first layer is updated with the rule $\mathcal{F}(\boldsymbol{\Theta}) = \theta_0 \mathcal{F}^0$ via feedback alignment, while the second layer uses $\mathcal{F}^{\text{eHebb}}(\boldsymbol{\Theta}) = \theta_0 \mathcal{F}^0 + \theta_2 \mathcal{F}^2$. Using

$\mathcal{F}^0$, information from $\boldsymbol{B}_{2,1}$ is communicated to $\boldsymbol{W}_{1,2}$ (①,③); meanwhile, the presence of $\mathcal{F}^2$ sets up a new channel to directly communicate information from $\boldsymbol{B}_{2,1}$ to $\boldsymbol{W}_{1,2}$ (②). The blue arrows depict information propagation through the forward and backward paths. The communications between feedback and feedforward pathways are represented with red arrows.

For a more precise, mathematical intuition of the effects that $\mathcal{F}^{\text{eHebb}}$ has on weights, we show in Supplementary Note 4 that, in a linear network model under reasonable approximating assumptions,

$$\mathbb{E}\left[\mathbf{e}_\ell \mathbf{e}_{\ell-1}^T \mid \boldsymbol{B}_{\ell,\ell-1}\right] \propto \boldsymbol{B}_{\ell,\ell-1}^T \qquad (9)$$

for layers, $\ell = 1, 2, \ldots, L-1$. Thus, the term $\mathbf{e}_\ell^T \mathbf{e}_{\ell-1}$ in $\mathcal{F}^{\text{eHebb}}$ pushes $\boldsymbol{W}_{\ell-1,\ell}$ toward the transpose of $\boldsymbol{B}_{\ell,\ell-1}$, resulting in faster alignment of the modulatory signals with the backprop algorithm's error vectors and more efficient learning.

### Oja's rule

Equation (7) proposes a plasticity rule to train deep networks using fixed feedback matrices. Above, we demonstrated that the Hebbian-style learning term improves the trained model's performance by improving the modulatory signals' alignments with the back-propagated analogs. Here, we look at the remaining plasticity term in Eq. (7): Oja's rule, a purely local learning rule that updates the weights based on its current state and the local activations in the forward path. To this end, we redefine the plasticity rule as a linear combination of the pseudo-gradient term and Oja's rule

$$\mathcal{F}^{\text{Oja}}(\boldsymbol{\Theta}) = -\theta_0 \mathbf{e}_\ell \mathbf{y}_{\ell-1}^T + \theta_9 (\mathbf{y}_\ell \mathbf{y}_{\ell-1}^T - (\mathbf{y}_\ell \mathbf{y}_\ell^T) \boldsymbol{W}_{\ell-1,\ell}). \qquad (10)$$

We initialize $\theta_0$ to $10^{-3}$ and $\theta_9$ to zero and employ Alg. 1 to optimize the set of meta-parameters $\boldsymbol{\Theta}$. Figure 7a, b illustrates that adding Oja's

rule to the pseudo-gradient term enhances the model's accuracy when backward connections are fixed. Figure 7c presents the angles between the teaching signals ensued by Eq. (10) and the corresponding back-propagated ones. While the accuracy and loss are significantly improved, contrary to expectations, Oja's rule does not substantially reduce the alignment angles (Fig. 7c). In fact, alignment angles are only slightly smaller when using Oja's rule compared to using pure FA, as seen by comparing Fig. 7c to Fig. 3d. This contrasts with alignment angles for $\mathcal{F}^{\text{eHebb}}$ and $\mathcal{F}^{\text{bio}}$, which are greatly reduced in deeper layers compared to $\mathcal{F}^{\text{Oja}}$ (compare Fig. 7c to Fig. 5c and Supplementary Fig. S2c).

Inspecting Fig. 7 suggests that rather than helping to align the modulating signals, Oja's rule helps by entirely circumventing the backward path. Oja's rule implements a Hebbian learning rule subjected to an orthonormality constraint on the weights[18]. In Eq. (10), $\mathbf{y}_{\ell-1}$ and $\mathbf{y}_\ell$ denote post-nonlinearity activations (as stated in Eq. (2)), resulting in the $\mathcal{F}^9$ plasticity rule to implement a non-linear version of Oja's rule. When trained iteratively, this non-linear variation implements a recursive non-linear algorithm for Principal Component Analysis[27,28]. Previous studies on the convergence of Oja's rule have shown that for a compression layer, where $\dim(\mathbf{y}_{\ell-1}) > \dim(\mathbf{y}_\ell)$, rows of the weight matrix $(\boldsymbol{W}_{\ell-1,\ell})_1, \ldots, (\boldsymbol{W}_{\ell-1,\ell})_{\dim(\mathbf{y}_\ell)}$ will tend to a rotated basis in the $\dim(\mathbf{y}_\ell)$−dimensional subspace spanned by the principal directions of the input $\mathbf{y}_{\ell-1}$[29].

We demonstrate that incorporating Oja's rule into Feedback Alignment improves feature map extraction in the forward path through unsupervised learning, despite $\mathcal{F}^{\text{Oja}}$ not recursively applying pure Oja's rule. By analyzing the continuous-time differential equation corresponding to the Oja's learning rule, Williams[29] and Oja[28] establish the stability limits for this rule. In a compression layer, the fixed point of Oja's rule is a stable solution if $\boldsymbol{W}_{\ell-1,\ell} \boldsymbol{W}_{\ell-1,\ell}^T = \boldsymbol{I}$. This conclusion can be used to derive a proximity measure[30–32] of the estimated $\boldsymbol{W}_{\ell-1,\ell}$ to a stable solution of Oja's rule in the presence of non-linear activations. The error

$$E_{\boldsymbol{W}} = |\mathbf{z}_\ell - \boldsymbol{W}_{\ell-1,\ell} \bar{\mathbf{y}}_{\ell-1}|_2^2, \qquad (11)$$

where

$$\bar{\mathbf{y}}_{\ell-1} = \boldsymbol{W}_{\ell-1,\ell}^T \sigma(\mathbf{z}_\ell), \qquad (12)$$

can define this measure. Figure 8 studies this orthonormality measure in models trained with different plasticity rules. Results show that using Oja's rule will render the weight matrices increasingly orthonormal, reducing the correlation in weight rows and improving the

### Table 1 | Effect of the Hebbian-like error learning rule $\mathcal{F}^{\text{eHebb}}$ on the alignment of the modulating signals $\alpha_\ell$ for different layers

| $\mathcal{F}^0$ | $\mathcal{F}^{\text{eHebb}}$ | $\alpha_0$ | $\alpha_1$ | $\alpha_2$ |
|---|---|---|---|---|
| $\boldsymbol{W}_{0,1}, \boldsymbol{W}_{1,2}, \boldsymbol{W}_{2,3}$ | – | 89.89 | 76.69 | 82.04 |
| $\boldsymbol{W}_{0,1}, \boldsymbol{W}_{2,3}$ | $\boldsymbol{W}_{1,2}$ | 89.95 | 59.95 | 72.14 |
| $\boldsymbol{W}_{0,1}, \boldsymbol{W}_{1,2}$ | $\boldsymbol{W}_{2,3}$ | 90.03 | 75.18 | 29.02 |
| $\boldsymbol{W}_{2,3}$ | $\boldsymbol{W}_{0,1}, \boldsymbol{W}_{1,2}$ | 75.29 | 61.23 | 72.56 |
| $\boldsymbol{W}_{0,1}$ | $\boldsymbol{W}_{1,2}, \boldsymbol{W}_{2,3}$ | 90.2 | 49.4 | 27.9 |
| $\boldsymbol{W}_{1,2}$ | $\boldsymbol{W}_{0,1}, \boldsymbol{W}_{2,3}$ | 84.86 | 74.25 | 30.33 |
| – | $\boldsymbol{W}_{0,1}, \boldsymbol{W}_{1,2}, \boldsymbol{W}_{2,3}$ | 77.93 | 49.93 | 28.4 |

The leftmost column includes the parameters updated using $\mathcal{F}^0$ with feedback alignment, and the next column indicates layers trained with $\mathcal{F}^{\text{eHebb}}$ (Eq. (8)). Angles $\alpha_\ell$ represent the alignment between the modulatory signal $\mathbf{e}_\ell$ and the backpropagated counterpart $\mathbf{e}_\ell^{\text{BP}}$ at each layer (in degrees). Since $\mathbf{e}_0$ is a synthetic error, the effect of the $\mathcal{F}^{\text{eHebb}}$ on $\boldsymbol{W}_{0,1}$ alone has been excluded. The model is trained for 500 episodes, and the computed angles are averaged after a burn-in period of 100 episodes.

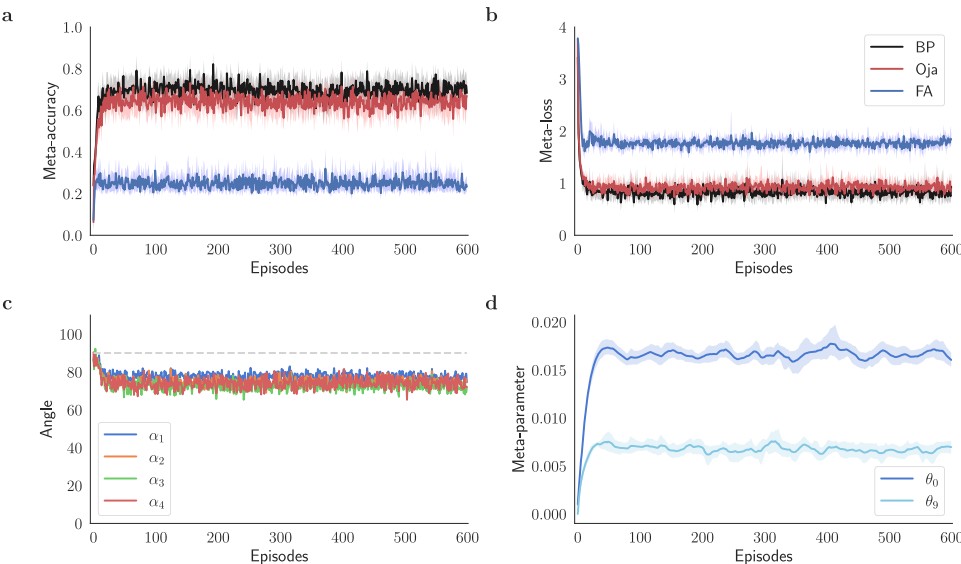

**Fig. 7 | Oja's rule helps with the slow learning introduced by random feedback connections.** Performance of the meta-learning model (Alg. 1) in training a 5-layer classification network using $\mathcal{F}^{\mathrm{Oja}}$ (Oja; Eq. (10)) through fixed backward connections. **a** Meta-accuracy and **b** meta-loss of $\mathcal{F}^{\mathrm{Oja}}$ compared to $\mathcal{F}^0$ learning rule via feedback alignment (FA) and backprop (BP). Although the modulatory signals of

$\mathcal{F}^{\mathrm{Oja}}$ are not much backprop-like, there is still substantial improvement in the model's performance, as evident in panels **a** and **b**. **c** Alignment angles $\alpha_\ell$ of modulating signals of $\mathcal{F}^{\mathrm{Oja}}$ with backprop's teaching signals (in degrees). **d** Evolution of the plasticity meta-parameters $\Theta = \{\theta_0, \theta_9\}$. The meta-learner converges in about 50 episodes.

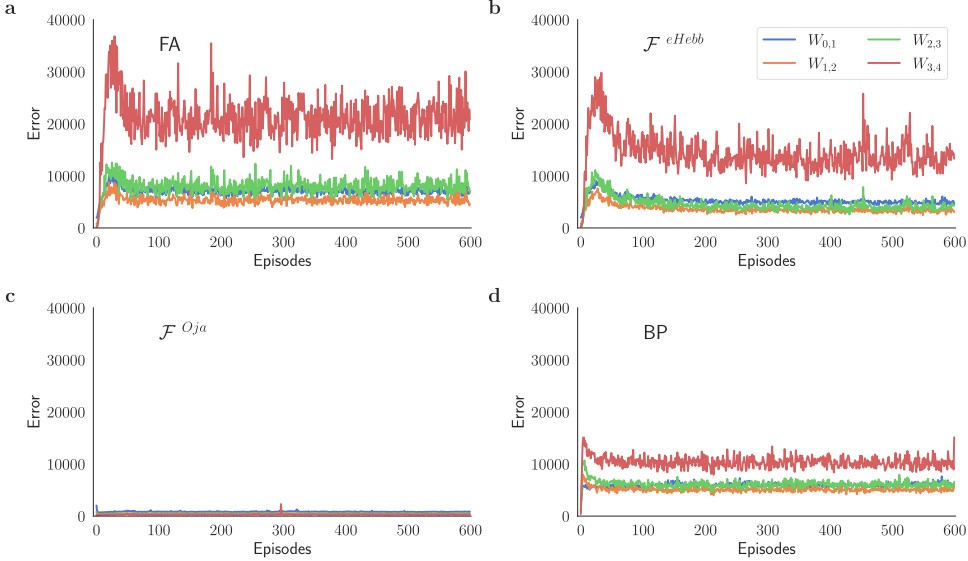

**Fig. 8 | Oja's rule substantially improves the orthonormality of the weight matrices.** Orthonormality error $E_W$ (Eq. (11)) computed for different plasticity rules. The error is measured at the end of each meta-optimization loop (episode) for different layers of a 5-layer deep network. The model is trained using **a** $\mathcal{F}^0$ via feedback alignment (FA), **b** $\mathcal{F}^{\mathrm{eHebb}}$, **c** $\mathcal{F}^{\mathrm{Oja}}$, and **d** backprop (BP). Results from panel

**c**, along with Fig. 7c, demonstrate the role of Oja's rule in improving the model's performance through improved feature extraction. All panels present the mean error over multiple trials (see Methods). In this comparison, the last layer has been excluded.

feature extraction in these layers. These findings indicate that introducing Oja's rule alone can help with the problem of slow learning caused by random feedback connections (see Supplementary Fig. S4).

The architecture of a classifier network includes initial layers that act as feature extractors, creating hidden representations for the final layer. This last layer, dubbed predictor, maps the hidden feature representations to the target class for the given input image. To improve the classifier's performance, a plasticity rule that enhances feature extraction in the earlier layers is beneficial. However, this rule has no grounds to positively impact the predictor layer's performance.

Despite this, for comprehensiveness, we also applied the plasticity rule $\mathcal{F}^{\mathrm{Oja}}$ to the final layer and found no detrimental effect on the model's performance.

In summary, rather than improving alignment, $\mathcal{F}^{\mathrm{Oja}}$ applied to hidden layers provides embeddings that facilitate more effective learning.

## Discussion
Despite the dominance of the backpropagation algorithm as the primary technique to train deep neural networks, its biological

plausibility remains a significant ground for contest[2,3]. In particular, the presence of feedback synaptic projections that are precisely symmetric to the forward projections is not biologically realistic. Previous work[5] showed that learning can be achieved without this symmetry using feedback connections that are randomly sampled, not tied to the forward path, and fixed throughout the training process. While a breakthrough, this method is susceptible to diminished performance when training deeper networks or using smaller batch sizes[6,7]. The latter is a challenge for online learning.

A recent body of work attempts to improve learning through asymmetric feedback connections. They either rewire fixed feedback connections, use plastic feedback connections that are updated through an auxiliary plasticity rule, or impose partial symmetry in the backward network[8,9,17,19,20]. Our work accelerates the learning process by enhancing the rules that govern neural plasticity while transmitting teaching signals through fixed connections. Our proposed rules for plasticity are based on biologically motivated learning principles, like Oja's rule, or have been inspired by them, such as the error-based Hebbian rule. A linear combination of these terms yields a parameterized learning rule. To overcome the arduous hand-tuning of these hyper-parameters, we use a meta-learning approach that systematically explores the pool of candidate plasticity rules. This approach consists of an inner loop that learns a task and an outer loop that updates the plasticity coefficients. The inner loop always starts from randomly initialized weights, so the model must learn to learn from scratch. Moreover, the inner loop learns from an online stream of training data, simulating real-time learning in the brain.

To assure interpretability of our meta-learned learning rule, we expressed the rule as a linear combination of individual plasticity terms, imposed an L1 penalty on the coefficients, and used meta-parameter sharing between all update rules. Many terms in the pool of plasticity rules can be redundant and employ identical or overlapping mechanisms but only differ in their efficiency, i.e., computational cost or the number of required learning iterations to operate. Employing an L1-penalized meta-loss decreases the count of plasticity terms that work in parallel. In addition, while sharing the same meta-parameters across layers may limit the model's freedom in learning, it is a vital component for discovering a global learning rule, leaving the door open to investigate the revealed terms.

Using this meta-learning approach, we discover two plasticity rules that accelerate learning through fixed feedback connections. The first, an error-based Hebbian rule, combines the errors of pre- and post-synaptic layers to update forward projecting weights. The second rule, known as Oja's rule, combines pre- and post-synaptic activations with the connection's current state to update weights. We investigated each plasticity rule, its underlying mechanism, and how it contributes to learning, revealing two distinct mechanisms behind them. First, the Hebbian-like error term improves performance by modifying the flow of information through the backward path. It introduces an auxiliary channel to communicate information about the backward connections to the forward weights. As a result, it accelerates learning by better aligning modulating signals with the ones transmitted through a symmetric feedback connection. Ultimately, the modified plasticity alters the training to resemble backpropagation. Unlike the Hebbian-like rule, Oja's rule does not directly affect the flow of the feedback signals. Instead, it acts only on the forward path, implementing an unsupervised learning scheme that extracts feature maps independently of the labels and loss. The updated weight rows approximate an orthonormal basis in the subspace spanned by PCA eigenvectors of the pre-synaptic activations[28]. The strengthened signal separation capabilities in the earlier layers improve predictions made by the output layer.

While synaptic plasticity in the brain is mediated by a vast array of biophysical processes, the changes to a single synaptic weight largely depend on the activity of its pre-synaptic and post-synaptic neurons

and the current weight, a property known as "local" plasticity. For the plasticity rules used in our study (with the exception of Oja's rule), weight updates depend on activations from a forward pass *and* error signals from a backward pass. Since these quantities were used to update the forward projecting weights, this raises the question of whether the plasticity rules are truly local. The answer to this question depends on the biological interpretation of the forward and backward passes.

Under one interpretation, separate populations of neurons encode the forward and backward passes, i.e., the neurons encoding $\mathbf{e}_\ell$ are distinct from those encoding $\mathbf{y}_\ell$. Under this interpretation, the plasticity rules used in this study are not strictly local.

Under another interpretation, forward activations and backward errors are represented by the same neural populations, i.e., the same neurons encode $\mathbf{e}_\ell$ and $\mathbf{y}_\ell$. Under this interpretation, all of the plasticity rules used in this study are local. There are several models for how this multiplexing of forward and backward signals could be achieved (see ref. [2] for a review). For example, activations and errors could be represented at separate points in time by the same neurons.

Alternatively, recent work hypothesizes that activations and errors are encoded separately in the basal and apical dendrites of the same cortical pyramidal neurons[33]. Along similar lines, a growing body of work posits that activations and errors are multiplexed by the distinction between bursts and single action potentials, which are communicated separately by synaptic projections onto the soma versus apical dendrites of pyramidal neurons[34–36]. The dependence of synaptic plasticity on the morphological site of the synaptic contact and on the type of spiking (bursts versus individual spikes) is well established in experiments[37–41]. Under these models, established biophysical properties of cortical synapses can produce plasticity rules like ours that multiplex forward and backward propagating information to update weights. Networks in[36] rely on weight decay to approximately align forward and backward weights[11], while some networks in[33] rely on random feedback alignment. Hence, our meta-learned plasticity rules could improve learning in those models.

Our meta-learning approach isolated three plasticity terms: a backprop-like rule ($\mathcal{F}^0$), Oja's rule[18] ($\mathcal{F}^9$), and a rule we refer to as eHebb ($\mathcal{F}^2$). Possible biological implementations of Oja's rule and the backprop-like rule have been studied in great depth in previous work[2,3,33,36]. The eHebb rule could be implemented in a similar way to the backprop-like rule. For example, under the model in ref. [36], eHebb would change synaptic weights in response to the co-occurrence of pre- and post-synaptic bursts. Plasticity is strongly mediated by firing rates and intracellular calcium[42,43], both of which are elevated during bursts.

As the eHebb's mechanism tends to align modulating signals with the symmetric counterparts, its performance may at best match that of backprop. However, as Oja's rule does not aim to imitate backprop, its performance is not bounded by that of backprop, and hence it can also be used to enhance learning in symmetric feedback models. For instance, we realized that adding Oja's plasticity rule to the gradient-based learning term accelerates learning for poorly initialized networks. This observation explains why the improved performance in the fixed feedback model may outperform learning in the symmetric case. A similar concept was used in the earlier works to initialize internal representations of the neural networks[32]. However, that work used weights preprocessed by Oja's rule to start gradient-based learning rather than using both terms simultaneously as the plasticity rule. Hence, our results demonstrate the utility of the proposed meta-learning approach as a tool for combining different learning terms as a single parameterized learning rule.

We used meta-learning to find plasticity rules that can learn effectively under the biologically relevant setting where forward and backward weights are not explicitly aligned. But our meta-learning technique can be applied more broadly to identify plasticity rules that

overcome other biological constraints in various contexts and models. For instance, our study only focused on plasticity in forward connections; however, backward projections in the brain can also exhibit plasticity. Our meta-learning approach can be extended to discover plasticity rules for backward connections in such settings. Another interesting future direction is to meta-learn the architecture of the feedback pathways instead of (or in addition to) the plasticity parameters. That is, to simultaneously provide both direct[8] and regular[5] feedback pathways and allow the meta-learning algorithm to pick the most efficient path to carry the teaching signals to each layer.

In another direction, our meta-parameter sharing approach could be partially relaxed without learning a new plasticity rule for each connection. For example, one could consider a network with several neural populations and a shared plasticity rule for each pair of populations. This approach could help understand the role of distinct neuron types and populations in biological circuits.

We focused on meta-learning biologically plausible plasticity rules, but our approach can also be applied to discover learning rules that satisfy other constraints or optimize other meta-loss functions. For example, the approach can be used to find learning rules that can be implemented in non-standard hardware like neuromorphic chips or optical networks, or to discover learning rules that minimize energy consumption or other factors.

In summary, we developed and tested a meta-learning approach designed to produce simple, interpretable plasticity rules that can effectively learn on new data. First, using randomly initialized weights on each iteration of the outer loop (instead of meta-learning the initialization) and using online learning in our inner loop encouraged plasticity rules that can perform online learning from scratch. Secondly, meta-parameter sharing yielded a vastly smaller set of learned plasticity rules compared to learning a plasticity rule for each synapse. Finally, an L1 penalty on plasticity coefficients promoted sparsity within the learning rule, ultimately yielding a small set of plasticity terms that are more readily interpreted. Our results demonstrate the utility of this approach for discovering and interpreting plasticity rules. Taken together, our work opens new avenues to the application of meta-learning for discovering interpretable plasticity rules that satisfy biological or other constraints.

## Methods
### Models
Figure 1 performs a 10-way classification on the MNIST dataset, with images resized to $28 \times 28$ dimensions. The model is trained online, processing one data point per iteration (batch size one) for a single epoch. The model is a 5-layer fully connected neural network with dimensions 784-170-130-100-70-47. Hidden layers use the softplus activation function

$$\sigma(\mathbf{z}_\ell) = \frac{1}{\beta}\log(1 + \exp(\beta \mathbf{z}_\ell)), \tag{13}$$

with $\beta = 10$. The output layer uses the softmax activation function. Figures 3–5 and 7–8 perform 5-way classification on the EMNIST dataset. During adaptation, the network is trained for one episode, with a batch size of one. These figures use the same architecture as Fig. 1. For Table 1, the model conducts a 5-way classification on the EMNIST dataset with an image size of $28 \times 28$. The model is a 3-layer fully connected neural network with dimensions 784-130-70-47. Like the rest of the paper, hidden layers use softplus non-linearity with $\beta = 10$, while the output layer uses softmax.

In the fixed feedback pathway problem, the weights and feedback connections are initially set to random values that differ from each other. Both symmetric and fixed feedback models utilize the Xavier method[44] to re-initialize forward and backward connections at the start of each meta-learning episode.

In Figs. 4, 5, 7, and 8, and Table 1, we set the initial value for the learning rate $\theta_0$ of the term $\mathcal{F}^0$ to $10^{-3}$ and set all other hyperparameters to zero.

All plots depict the mean outcome over 20 trials, each with different initial weights and feedback matrices. The shaded region in the loss, accuracy, and meta-parameters plots illustrates the 98% confidence interval, determined through bootstrapping across trials with 500 bootstrapped samples.

### Candidate learning terms
Equation (7) presented a plasticity rule that improves the model's performance in the presence of fixed random feedback connections. We employed the meta-learning framework described in Alg. 1 to explore a set of local learning rules to discover such a plasticity term. This set of terms is defined as

$$\mathcal{F}^0 = -\mathbf{e}_\ell \mathbf{y}_{\ell-1}^T, \tag{14}$$

$$\mathcal{F}^1 = -\mathbf{y}_\ell \mathbf{e}_{\ell-1}^T, \tag{15}$$

$$\mathcal{F}^2 = -\mathbf{e}_\ell \mathbf{e}_{\ell-1}^T, \tag{16}$$

$$\mathcal{F}^3 = -\mathbf{W}_{\ell-1,\ell}, \tag{17}$$

$$\mathcal{F}^4 = -\mathbf{1}_\ell \mathbf{e}_{\ell-1}^T, \tag{18}$$

$$\mathcal{F}^5 = -\mathbf{e}_\ell \mathbf{1}_\ell^T \mathbf{y}_\ell \mathbf{y}_{\ell-1}^T, \tag{19}$$

$$\mathcal{F}^6 = -\mathbf{y}_\ell \mathbf{y}_\ell^T \mathbf{W}_{\ell-1,\ell} \mathbf{e}_{\ell-1} \mathbf{e}_{\ell-1}^T, \tag{20}$$

$$\mathcal{F}^7 = -\mathbf{e}_\ell \mathbf{y}_\ell^T \mathbf{W}_{\ell-1,\ell} \mathbf{e}_{\ell-1} \mathbf{y}_{\ell-1}^T, \tag{21}$$

$$\mathcal{F}^8 = -\mathbf{y}_\ell \mathbf{y}_{\ell-1}^T \mathbf{W}_{\ell-1,\ell}^T \mathbf{e}_\ell \mathbf{e}_{\ell-1}^T, \tag{22}$$

$$\mathcal{F}^9 = (\mathbf{y}_\ell \mathbf{y}_{\ell-1}^T - (\mathbf{y}_\ell \mathbf{y}_\ell^T) \mathbf{W}_{\ell-1,\ell}). \tag{23}$$

The rules above are local in the sense that the updates to the $j, k$th entry of $\mathbf{W}_{\ell-1,\ell}$ depend only on the $k$th entry of $\mathbf{e}_{\ell-1}$ and $\mathbf{y}_{\ell-1}$, the $j$th entry of $\mathbf{y}_\ell$ and $\mathbf{e}_\ell$, and the $j, k$th entry of $\mathbf{W}_{\ell-1,\ell}$. This notion of locality assumes that errors and activations are encoded in the same neurons (see Discussion). Even under this constraint of locality, there is an unlimited number of possible plasticity rules to choose from. To form the list above, we first considered all quadratic combinations of activations and errors except we omitted pure Hebbian plasticity ($\mathbf{y}_\ell \mathbf{y}_{\ell-1}^T$) because we found that it leads to unstable network dynamics (a blowup of activations). Instead, we replaced it with Oja's rule $\mathcal{F}^9$, which adds a stabilizing term onto pure Hebbian plasticity. Additional terms were added to test the viability of higher order plasticity terms.

Computing the learning terms $\mathcal{F}^1, \mathcal{F}^2, \mathcal{F}^4, \mathcal{F}^6, \mathcal{F}^7$, and $\mathcal{F}^8$ requires a pre-synaptic error term. In order to update the weights in the first layer $\mathbf{W}_{0,1}$, where there is no pre-synaptic error, we define a synthetic error $\mathbf{e}_0$ using Eq. (3) and the activation function in Eq. (13), such that

$$\mathbf{e}_0 := \mathbf{B}_{1,0} \mathbf{e}_1 \odot (1 - \exp(-\beta \mathbf{y}_0)). \tag{24}$$

### Meta-training
We presented a meta-learning framework for swiftly exploring a pool of plasticity terms and uncovering combinations that exceed the performance of the existing plasticity rule. We demonstrate this by

training a classifier network, which performs a 5-way classification on $28 \times 28$ images. The cross-entropy function evaluates the loss in the adaptation loop, whereas the meta-loss is determined by Eq. (6). While, in principle, any optimization algorithm, such as evolutionary methods, can be used to optimize $\Theta$, the algorithm presented in Alg. 1 uses ADAM[45], a gradient-based optimization technique, with a meta-learning rate of $10^{-3}$.

In the meta-optimization phase, this gradient-based optimizer differentiates through the unrolled computational graph of the adaptation phase. Thus, the non-linear layers are double differentiated, once to compute $e_L$ and a second time by the meta-optimizer. This arrangement will only allow a two-times differentiable non-linear layer, which prohibits using the Rectified Linear Unit, ReLU, as the activation function $\sigma$. Instead, we use the softplus function (Eq. (13)), a continuous, twice-differentiable approximation of the ReLU function. In Eq. (13), parameter $\beta$ controls the smoothness of the function. Furthermore, the L1 norm used in the meta-loss (Eq. (6)), defined by the absolute value function, is not continuously differentiable at every point. However, it is commonly used in deep learning in conjunction with stochastic gradient descent (SGD)[46]. In PyTorch and other deep learning frameworks, the derivative of the absolute value function is typically defined as zero at zero.

In the present examples, each task contains $M = 5$ labels. Consequently, assembling a diverse set of 5-way classification tasks requires a database with a large number of classes. Thus, databases such as MNIST[47], which only has ten classes, are unsuitable for proper meta-training. On the other hand, in each episode, the classifier $f_W$ is reinitialized with random weights $W$. Therefore, the task should contain enough data points per class to train $f_W$ adequately. Hence, databases such as Omniglot[48] with only 20 data points per character designed for few-shot learning (e.g., with meta-optimized $W$) are impractical in the present framework. In the current work, meta-training tasks are made from the EMNIST database[49]. This database contains 47 classes, making it a good candidate for the meta-learning framework. Each task contains $K = 50$ training and $Q = 10$ query data points per class.

Notably, the use of $K = 50$ training data per class with $M = 5$ classes in each episode means that the meta-learned plasticity rule needs to train a randomly initialized network with only 250 training data points. Hence, our models are in a low data regime without the benefit of pre-trained weights that are often used for few-shot learning.

## Data availability
In this study, the EMNIST database[49] was used for meta-learning experiments. This database is publicly accessible at https://doi.org/10.1109/IJCNN.2017.7966217. Additional benchmarking was done using the MNIST dataset[47] and the FashionMNIST dataset[50]. These datasets can be found at http://yann.lecun.com/exdb/mnist and https://github.com/zalandoresearch/fashion-mnist, respectively. Source data are provided with this paper.

## Code availability
The PyTorch-based implementation and script files used to generate the results in this paper can be accessed at https://github.com/NeuralDynamicsAndComputing/MetaLearning-Plasticity[51].

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

## Acknowledgements
The authors thank Ashok Litwin-Kumar, Jack Lindsey, Claudia Clopath, and Klara Kaleb for their helpful discussions. This work was supported by Air Force Office of Scientific Research grant FA9550-21-1-0223 (N.S.T., R.R.); and by National Science Foundation grants NSF-DBI-1707400 (N.S.T., R.R.), and NSF-DMS-1654268 (R.R.).

## Author contributions
N.S.T. and R.R. conceived and designed the project. N.S.T. performed numerical experiments and generated figures. N.S.T. and R.R. wrote the manuscript.

## Competing interests
The authors declare no competing interests.
