## [Peer review file · Nature Communications]

REVIEWER COMMENTS

Reviewer #1 (Remarks to the Author):

The introduction is very clear, and it useful to understand a better update rule under random backwards update connections. This rule is obtained by meta-learning weights on existing update rules, with an L1 regularizer, to subselect amongst these rules.

I have two key concerns, that I believe can be easily addressed. Following that I have a longer set of comments to hopefully help improve clarity in the work.

The biggest omission here is a better connection to other work on learning update rules. They are not necessarily focused on biological plausibility (though some are), nor on improving random feedback alignment. But they are trying to understand alternative update rules. Examples include, among many others:

“Learning to learn by gradient descent by gradient descent” Andrychowicz et al.

“Meta-Learning Update Rules for Unsupervised Representation Learning”, Metz et al.

It would be useful for this work to summarize the insights from this literature, particularly with respect to the problem setting considered in this work.

Second, the paper states the following contribution:

“In this study, we develop a novel meta-plasticity approach to discover interpretable, biologically plausible plasticity rules that improve online learning performance with fixed random feedback connections. The resulting plasticity rules show improved online training of deep models in the low data regime. Our results highlight the potential of meta-plasticity to discover effective, interpretable learning rules satisfying biological constraints. ”

The implication is that learning a linear combination on a set of existing plasticity rules can help identify good, biologically plausible update rules. (Linearity is key for interpretability) But this is not explicitly stated up front, nor are the ramifications discussed. Does this mean that you believe a linear combination of existing rules will get us there? Or that we already have the single rule that works well, but have not had a way to find it across problems? And is one of the conclusions from this work that Oja’s rule plus random feedback connections is a pretty good plasticity rule? I think one of the critical contributions here is choosing to parameterize the update rule as this linear combination (rather than meta-plasticity, which has been previously investigated), so really focusing on this and explaining why it is critical would help clarify the key contribution here. For me, some of the insights in the work (about Hebbian learning and Oja’s rule) were more interesting than the meta-plasticity approach. Your simple approach helped get those insights, so of course is important.

Otherwise, this paper is nice. Below is a list of hopefully constructive comments, expressed in order as reading the paper:

I was a tad confused about some of the choices, until I got to Section 2.2.2. Once I understood that $\text{mathcal{F}}$ (the update function) was parameterized as a linear combination of existing update rules, then the objective function (including the l_1 norm) was much more clear. I wonder if it is possible to propose this parameterized form earlier, and also contrast it with the less interpretable rule mentioned in the introduction, by Lindsey et al. How did they parameterize the function?

And further, how does your approach compare to theirs? It is ok for it to perform worse, since you restrict your form more for interpretability. However, such a comparison would be illuminating.

As a minor comment, you use the gradient of the loss plus the l_1 regularizer. But, the l_1 regularizer is not differentiable everywhere, and proximal updates can be better. Did you consider this? You could acknowledge that you have a subgradient, and also potentially reference work that says SGD for the l_1 is reasonable effective (whereas when using GD, I have generally found proximal methods to help a lot).

In Figure 4, the coefficients are negative for two of the terms used! That says that you do the opposite of those plasticity update rules (Oja's rule and the Hebbian term). How can that be? Is it because those rules are usually added, and you are subtracting? If so, it would be useful to make them all the same sign (change the rules so they are all supposed to be usually added or subtracted) to improve interpretability.

A minor naming convention. Somehow when I first read meta-plasticity, I assumed this would mean learning a rule that maintained plasticity. Instead, you are learning an update rule. NNs, though, are known to lose the ability to learn, and so meta-learning could be used to avoid this issue.

Pg 10 "Despite this, the alignment angles of FeHebb (Fig. S2) are superior to Fbio's, ". The word superior is a bit confusing, since here performance is better for Fbio. Instead, I think you mean: are better aligned with the backprop direction. It is not clear (nor likely true) that backprop is the best direction.

Figure 6 was a bit vague. I wonder if this could be augmented to help better understand the description in the text. I didn't much follow it. Maybe adding the update equations there might help understand.

Table 1 is confusing. What are the α_0 , α_1 and α_2 for? Just for the Hebbian angles, or somehow a ratio between the angles for Hebbian versus random alignment?

How did you pick the 9 possible plasticity rules? Are they all biologically plausible rules?

The text states that adding Oja's rule improves performance, but not alignment. But, it seems to me that alignment for Fbio and when just using Oja's (plus the gradient term) is not *too* different. Is the implication that Fbio has aligned well, but Oja's rule has not?

Finally, you did a nice thorough analysis on this dataset. That is enough. But it would also be interesting to see if this behavior of Oja's rule extends to other datasets, or if we see that other rules are much more effective there. This is not a criticism so much as: your work has intrigued me and I want to know more.

Minor comments:

1. Eq 6 should have $L_{\text{meta}}(\Theta)$, so it is clear that the objective function is for Θ .
2. Figure 2 will also be more useful once you introduce the functional form for $\text{mathcal{F}}$ earlier. This figure was not useful to me until I read 2.2.2.

Reviewer #2 (Remarks to the Author):

Shervani-Tabar and Rosenbaum explore possible learning rules with the aim to boost the performance of a feedforward network on a classification task that relies on random feedback alignment to backpropagate errors through the network. The authors propose 10 mathematical terms and use a meta-learning approach to show that three of the explored terms are sufficient to increase the accuracy of the network to similar levels as when it is trained with a backprop algorithm.

The manuscript is well written and the results are interesting, shining a light on how the brain (with biological constraints) might implement algorithms that are similar to backprop. I appreciate the analysis of the individual plasticity terms that the authors find with their meta-learning approach. I also consider it relevant because it explores possible improvements to the random alignment algorithm. It is unclear to me, however, how pre- and post-synaptic errors would be encoded and transmitted to specific synapses so that the meta-learned rules are biologically plausible, as claimed by the authors. I have other specific points which I elaborate below on (major and minor points) which I think are important to be addressed by the authors.

Major points

- 1) It is unclear whether the rules discovered by the author's meta-learning approach are indeed biologically plausible. Apart from Oja's rule, which is solely composed of pre- and post-synaptic activities as well as the connection weight, the other two "bio" terms (referred to as "eHebb") use pre- and post-synaptic errors that are presumably computed by a backward pathway. It seems to me that this makes the rule nonlocal and the concept of biological plausibility may be lost. In this vein, it would be interesting to compare the author's model to biophysical implementation of backprop in Sacramento et al. (NeurIPS 2018). Could the authors elaborate how the learning rule found by their meta-learning approach could be implemented with biological constraints?

2) The meta-learning algorithm (Algorithm 1) does not seem to be entirely novel as described by the authors in the abstract: "In this study, we develop a novel meta-plasticity approach to discover interpretable, biologically plausible plasticity rules.." (see, e.g., Confraveux et al. 2020). In my view, what seems to be novel is the set of terms used in the search space of the meta-learning algorithm (Section 4.2). Could the authors elaborate on the precise novelty in the algorithm?

3) The authors argue that imposing an L1 penalty on the plasticity coefficients encourages their algorithm to learn a plasticity rule with fewer terms. This is a valid point, however, it should be confirmed with, e.g., comparing it to using L2 (or maybe not using a penalty term if that's possible). Is it truly a key feature of the author's approach or would the results be similar with a different norm? I might be missing the point here because it sounds like a technical detail rather than a key feature.

4) For the candidate learning terms (Section 4.2), all terms are of increasing complexity, yet "simple" multiplications of different variables apart from the last term, F9, which is Oja's rule. It is not clear to me why a standard Hebb term, $y_l * y_{l-1}^T$, as well as the Oja's penalty term, $y_l * y_{l-1}^T * W_{l-1,l}$, were not used as independent candidate terms, which would make the meta-learning search more independent/robust. Could the authors elaborate on why they have chosen to explicitly use Oja's rule? Additionally, what would happen if Oja's rule was separated into two terms, a purely Hebbian and the penalty term?

5) The meta-learning algorithm finds 3 terms as relevant to solve the classification task chosen by the authors (as shown in fig. 4), however, one of these terms does not seem to be needed if I interpreted fig. 7 correctly. This raises the question of whether the outer (meta-learning) loop reached a global minimum after 500 episodes (fig. 4), given that the error is similar to backprop with (fig. 4) and without F2 (fig. 7). Could the authors please clarify whether this interpretation of fig. 7 makes sense? I would like to suggest running longer sims (with more episodes) to make sure that a minimum has been reached in the outer loop.

Minor points

6) The term "meta-plasticity" is usually used in neuroscience to refer to the "plasticity of synaptic plasticity" (see, e.g., Abraham and Bear, 1996), which differs in meaning to "meta-learning", usually used in machine learning for automatic algorithm learning (well explained by the authors in page 5, line 143). For example, "meta-plasticity" is used in page 2, line 68; page 3, line 78; page 3, line 83; and other places, where it means (as far as I understand) "meta-learning".

7) Page 3, line 83: "We use meta-plasticity to learn a parameterized plasticity rule based on a combination of biologically motivated rules." It is not clear to me how these terms (outlined in section 4.2) are biologically motivated (please see point 1).

8) Page 3, line 86-91: The first two key features presented by the authors seem to have considerable overlap with Confraveux et al., 2020. If that's the case, I would like to suggest adding a citation there and clarifying this in the text.

9) Page 3, line 92: It is not clear to me how relevant this feature is for the results found by the authors. Would the method not work using L2? (please see point 3).

10) Page 3, line 95: Previous work on meta-learning plasticity rules (e.g., Confavreux et al., 2020; Lindsey and Litwin-Kumar, 2020) have always considered "online rules". Could the authors elaborate on which would be the other possible type of implementation apart from the online learning in their setting? Why is this a key feature of the authors' approach?

11) Page 4: The authors explain well the backprop algorithm, but I find it confusing to have Eq. 4 defining the matrix B as the transpose of W , given that B is also used for the random feedback alignment algorithm. The authors later in the page define superscripts "FA" and "BP" for error computed with random feedback alignment and backprop, respectively, but these superscripts are not used in, e.g., page 6, when they define the meta-learning algorithm. I would like to suggest to the authors to always use a superscript in both error and matrix B (starting in Eq. 3) to clearly state when they are referring to the transpose of W or a fixed matrix.

12) Page 4, line 131: "Alternative approaches have proposed a direct feedback pathway... but we find that this also does not achieve backprop performance" Could the authors elaborate? I think the manuscript would be more clear with a longer explanation of these findings here.

13) Page 5, fig 1a: How is the matrix B generated for FA? The "speed" by which it reaches high accuracy is always the same for different initialisations of B ?

14) Page 7, fig 2: In the schematics (middle of figure), it looks like the activation variables, y_i and y_j , are computed separately from the errors, e_i and e_j . The implementation in a simulation is straightforward, but the biological implementation is not (as intuitively seen in the schematics). Could the authors elaborate on how this type of plasticity would be implemented in a biological setting? I would like to suggest adding such an implementational hypothesis to this figure.

15) Page 7, line 180: I would like to suggest explicitly describing panels a to c of figure 3 in the main text to help the readers.

16) Page 9, (no line numbers on this page): I would like to suggest explicitly describing panels a to c of figure 4 in the main text to help the readers.

17) Page 9, fig 4d: It's not clear from the figure whether the system (outer loop) reached a global minimum after 500 meta-learning iterations (episodes). Please see point 5.

18) Page 15, line 295: "The proposed plasticity rules draw upon biological learning rules, including Hebbian plasticity and Oja's rule, ..." To the best of my knowledge, Oja's rule is a type of Hebbian plasticity rule. I would like to suggest clarifying it in the Discussion section.

19) Page 15, line 296: Continuing last point; "... to combine information locally from their immediate neighbors, ..." It's not clear to me what the authors mean by "neighbors" in this sentence. The same term is used later in page 16, line 307, without being clear to me either what the term means.

20) Page 16, line 306: "The first one uses a Hebbian-style learning term ..." In the previous page the authors refer to it as "Hebbian plasticity" rather than "Hebbian-style learning". I find these terms a bit misleading because, as far as I know, Oja's rule is an implementation of Hebbian learning and it could be referred to as Hebbian-style learning, however, the authors state that a learning rule with error terms is Hebbian. I would like to suggest clarifying these statements.

21) Page 16, line 335: "... interpretable learning rules satisfying biological constraints ..." These constraints are not entirely clear to me. The next sentence states that "... we only considered plasticity in the forward connections..." However, this is, in my view, not a strong biological constraint, such as, e.g., plasticity being local. The manuscript's clarity would benefit from an introduction of what biological constraints are being imposed to arrive at the terms introduced in section 4.2.

22) Page 16, line 374: "We focused on metal-learning biologically motivated plasticity rules, ..." Similar to the point above, the manuscript's clarity would benefit from a more detailed explanation of how each term (or rule) follows a specific biological constraint.

23) Page 26, line 529: The "big" parentheses to the right of " i,k " and " j,k " should be moved to the right so that the sum over k and k' also takes into account the error terms.

Reviewer #3 (Remarks to the Author):

This paper is interesting, well-written, and makes a valuable contribution. I think it should be published once the issues I raise are addressed (either by explaining why they should not be addressed, or, more likely, by modifying the manuscript to address them).

I only have one major concern. Throughout the paper in every result there is no discussion of between-run variation. To be scientifically sound, one should do the same experiment multiple times with different random factors (e.g. different initial weights) and then have plots that show intra- and inter-treatment variation. Was each treatment only run once? If so, how can we conclude what is noise vs. signal? Similarly, there should be statistics performed (and proper ones; meaning not assuming normally distributed data unless you first do the proper tests to confirm normality) to show whether the different treatments are statistically significantly different. I recommend redoing the experiments and plotting/reporting bootstrapped confidence intervals, doing statistical tests (e.g. Mann-Whitney tests), etc.

More specific issues

- Lines 70-71. Do these papers not have controls that only optimize the initial weights, but without plasticity rules? If so, wouldn't the performance gap between the two tell you the affect/benefit of the learned plasticity rules?

- 261-268: It is not clear in these lines how much of the phenomena you describe were known ahead of time (and, if so, which citations support them; there seems to be a lack of sufficient citations given the claims being summarized) vs. new claims/phenomena you are making (in which case supporting data/descriptions seems missing). I think most of these claims were previously known, meaning what is needed is clearer writing and more citations. On line 266, citation [26] is at an unconventional location in the sentence, making it difficult for me to figure out what claim that citation is supposed to support.

- Lines 277- 279: Confusing and seemingly contradictory. How is it both true that it is not appropriate to train the last layer this way, but that you did train it that way without any performance degradation (at least, I think that is what you are saying). Overall these lines are unclear. Can you explain more clearly what exactly you did?

Response to reviewers

We thank the reviewers and editor for their helpful comments that have improved the manuscript. Below, we have included a response to each of the reviewers' comments. Reviewer's comments are in **bold text**, our responses are in regular text, and text added to the manuscript is in **red**. Citation numbers in this document refer to the bibliography at the end of this document, not the main text.

Reviewer 1

The introduction is very clear, and it useful to understand a better update rule under random backwards update connections. This rule is obtained by meta-learning weights on existing update rules, with an l1 regularizer, to subselect amongst these rules.

I have two key concerns, that I believe can be easily addressed. Following that I have a longer set of comments to hopefully help improve clarity in the work.

Major

- 1. The biggest omission here is a better connection to other work on learning update rules. They are not necessarily focused on biological plausibility (though some are), nor on improving random feedback alignment. But they are trying to understand alternative update rules. Examples include, among many others:**

- **“Learning to learn by gradient descent by gradient descent” Andrychowicz et al.**
- **“Meta-Learning Update Rules for Unsupervised Representation Learning”, Metz et al.**

It would be useful for this work to summarize the insights from this literature, particularly with respect to the problem setting considered in this work.

We have modified the introduction to include a summary of the literature above (lines 77-79, 83-89). In addition, we have included the following related works (lines 90-95)

- “Meta-learning biologically plausible semi-supervised update rules” by Gu *et al.* [1]
- “Meta-learning bidirectional update rules.” by Sandler *et al.* [2]

More recent work includes Andrychowicz *et al.* [3], who parametrize the learning rule with a Recurrent Neural Network (RNN) and meta-learn weights of the RNN model. Using an RNN allows for training a dynamic update rule.

The scope of the meta-learning framework is beyond learning the forward pathway's plasticity rule. Meta-learning has given rise to unorthodox training models beyond the classic backward transmission of errors. For example, Metz *et al.* [4] used a meta-learning framework to learn a plasticity rule for unsupervised learning. They proposed to infer the teaching signals by meta-learning a network that projects forward activation units and the downstream feedback signal into backward hidden states. These hidden states are subsequently used to update the forward and backward weights via each pathway's meta-learned plasticity rule. Another related work

on semi-supervised learning [1] uses learnable auxiliary feedback and lateral connections to facilitate error propagation during training and meta-learns the plasticity rules to update these connections. Finally, Sandler *et al.* [2] reformulate the interactions between the forward and backward activations by defining parameterized update rules for both feedforward and feedback connections. Then, they yield new plasticity rules by meta-learning these hyperparameters.

2. **Second, the paper states the following contribution: “In this study, we develop a novel meta-plasticity approach to discover interpretable, biologically plausible plasticity rules that improve online learning performance with fixed random feedback connections. The resulting plasticity rules show improved online training of deep models in the low data regime. Our results highlight the potential of meta-plasticity to discover effective, interpretable learning rules satisfying biological constraints.”** The implication is that learning a linear combination on a set of existing plasticity rules can help identify good, biologically plausible update rules. (Linearity is key for interpretability) But this is not explicitly stated up front, nor are the ramifications discussed. Does this mean that you believe a linear combination of existing rules will get us there? Or that we already have the single rule that works well, but have not had a way to find it across problems? And is one of the conclusions from this work that Oja’s rule plus random feedback connections is a pretty good plasticity rule? I think one of the critical contributions here is choosing to parameterize the update rule as this linear combination (rather than meta-plasticity, which has been previously investigated), so really focusing on this and explaining why it is critical would help clarify the key contribution here. For me, some of the insights in the work (about Hebbian learning and Oja’s rule) were more interesting than the meta-plasticity approach. Your simple approach helped get those insights, so of course is important.

The reviewer is correct that our utilization of a linear combination of plasticity rules is key for the interpretability of our meta-learned rule. The combination of this linearity with an L1 penalized meta-loss and meta-parameter sharing allowed us to identify the dominating terms among a pool of learning terms and further study them (as performed in Section 2.2.2 of this work). While we mentioned out the L1 penalty and meta-parameter sharing in the text, we did not mention the importance of using a linear combination of individual plasticity rules. The reviewer also questions whether we believe that a linear combination of existing rules or a single rule is ideal. Our study is mostly agnostic to these two alternatives since our approach can meta-learn one or more rules. It should also be noted that many of the terms we included in our pool of candidate rules are not established plasticity rules. And a linear combination of plasticity rules is itself another plasticity rule (indeed, Oja’s rule is already a sum of two terms), so the question of whether we aim to learn a single existing rule or a combination of existing rules can become somewhat ambiguous. We added new text (lines 348-356) to the Discussion to address these comments and explicitly point out the importance of writing the learning rule as a linear combination of plasticity rules.

To assure interpretability of our meta-learned learning rule, we expressed the rule as a linear combination of individual plasticity terms, imposed an L1 penalty on the coefficients, and used meta-parameter sharing between all update rules. Many terms in the pool of plasticity rules can be redundant and employ identical or overlapping mechanisms but only differ in their efficiency, *i.e.*, computational cost or the number

of required learning iterations to operate. Employing an L1-penalized meta-loss decreases the count of plasticity terms that work in parallel. Additionally, while sharing the same meta-parameters across layers may limit the model’s freedom in learning, it is a vital component for discovering a global learning rule, leaving the door open to investigate the revealed terms.

We also emphasized the linear combination of plasticity rules when it is first introduced (lines 185-186).

Then, given a set of R candidate terms $\{\mathcal{F}^r\}_{0 \leq r \leq R-1}$, a parametrized plasticity rule is defined as a linear combination of individual plasticity terms.

The reviewer also asked whether a conclusion of our study is that Oja’s rule by itself is a good learning rule. In the context of our study, this should be interpreted to mean that Oja’s rule alone can help overcome slow learning introduced by random feedback connections. We do think this is a conclusion of our study and we added the following sentence to point this out more explicitly (lines 315-316).

These findings indicate that introducing Oja’s rule alone can help with the problem of slow learning caused by random feedback connections.

We have realized that not only does Oja’s rule benefit the random feedback model, but it also can be helpful to improve learning coupled with backprop. Therefore, it extends beyond the Feedback Alignment model. In addressing an upcoming comment (Minor-11), we show that this behavior of Oja’s rule also extends to the other datasets (See Supplementary material; Fig. S4).

Otherwise, this paper is nice. Below is a list of hopefully constructive comments, expressed in order as reading the paper:

Minor

- 1. I was a tad confused about some of the choices, until I got to Section 2.2.2. Once I understood that \mathcal{F} (the update function) was parameterized as a linear combination of existing update rules, then the objective function (including the L1 norm) was much more clear. I wonder if it is possible to propose this parameterized form earlier**

We thank the reviewer for pointing out the potentially confusing order in which terms were defined. To resolve this, we moved the definition and description of the update function’s parameterization ($\mathcal{F}(\Theta)$) earlier (in Section 2.2), prior to the definition of the meta-objective function (lines 185-195).

- 2. Also, contrast it with the less interpretable rule mentioned in the introduction, by Lindsey et al. How did they parameterize the function? And further, how does your approach compare to theirs? It is ok for it to perform worse, since you restrict your form more for interpretability. However, such a comparison would be illuminating.**

The work by Lindsey *et al.* [5] is hard to interpret because it does not limit meta-learning to the plasticity rule but also meta-learns initial weights and feedback connections.

Lindsey *et al.* proposed a two-step supervised Oja’s rule, where first, the activations are updated using

$$\mathbf{y}'_\ell = (1 - \theta'_\ell) \mathbf{y}_\ell + \theta'_\ell \mathbf{e}_\ell,$$

where \mathbf{e}_ℓ is the teaching signal propagated through directed feedback. Then, weights are updated by

$$\mathcal{F} = \boldsymbol{\theta}_2 \odot (\mathbf{y}'_\ell \mathbf{y}'_{\ell-1}{}^T - (\mathbf{y}'_\ell \mathbf{y}'_\ell{}^T) \mathbf{W}_{\ell-1, \ell}).$$

where θ'_ℓ is associated with each layer, $\boldsymbol{\theta}_2$ ’s elements are associated with each weight, and \odot is the element-wise product.

However, the problem with the interpretability of this work arises from the following differences. In the work of Lindsey *et al.*:

- Different layers and weights use different plasticity parameters. Lack of parameter-sharing prevents learning a general learning rule.
- Weight initializations are meta-learned. While we reinitialize the network weights to random values at the beginning of each episode, their work uses weights that are meta-learned in the previous episode as the initial weights. Learning the starting point in the parameter space accelerates the learning speed.
- Feedbacks are meta-learned. While we use fixed randomly sampled feedback connections at each episode, they use meta-learned feedback connections from the previous episode. Meta-optimizing feedback connections lead to improvement in error transmission efficiency.

These issues prevent studying the extent to which the learning rule has affected the improvements in the learning, as opposed to optimizing the feedback connections and initial weights. Also, it prevents generalizing the learned rule across layers. Further, it prevents generalizing the learned rule to a lifetime, as the update rule adapts to the meta-learned feedback and initial weights. We added a reference to Lindsey *et al.*’s work in Section 2.2 (line 180), where we discuss these issues.

3. **As a minor comment, you use the gradient of the loss plus the l1 regularizer. But, the l1 regularizer is not differentiable everywhere, and proximal updates can be better. Did you consider this? You could acknowledge that you have a subgradient, and also potentially reference work that says SGD for the l1 is reasonable effective (whereas when using GD, I have generally found proximal methods to help a lot).**

The reviewer is correct that the L1 penalty is not differentiable at zero. We used PyTorch automatic differentiation tools, which defines the derivative of the absolute value function as zero at this point. This convention fits the idea that L1 regularization does not penalize parameters with zero value. Moreover, exact zeros do not frequently arise in practice when using floating point arithmetic. The use of gradient-based learning with L1 norms is common in deep learning and, as pointed out by the reviewer, tends to work reasonably well when using SGD. To account for these comments, we added the following sentence with a reference to a textbook in which L1 regularization for deep learning is discussed (lines 498-502):

Furthermore, the L1 norm used in the meta-loss (Eq. 6), defined by the absolute value function, is not continuously differentiable at every point. However, it is

commonly used in deep learning in conjunction with stochastic gradient descent (SGD) [6]. In PyTorch and other deep learning frameworks, the derivative of the absolute value function is typically defined as zero at zero.

4. **In Figure 4, the coefficients are negative for two of the terms used! That says that you do the opposite of those plasticity update rules (Oja’s rule and the Hebbian term). How can that be? Is it because those rules are usually added, and you are subtracting? If so, it would be useful to make them all the same sign (change the rules so they are all supposed to be usually added or subtracted) to improve interpretability.**

We thank the reviewer for pointing out this issue, which could make it more difficult to interpret some Figures. Note that several of our plasticity terms are not previously studied rules so they do not have an established sign convention. For those rules that have an established sign convention, we modified the definition so that a positive coefficient reflects the more common version of the rule (Eq. 7, Eq. 9, and Section 4.2). The signs of some curves in Figs. 4d, 7d, and S2d are modified accordingly.

5. **A minor naming convention. Somehow when I first read meta-plasticity, I assumed this would mean learning a rule that maintained plasticity. Instead, you are learning an update rule. NNs, though, are known to lose the ability to learn, and so meta-learning could be used to avoid this issue.**

The reviewer is correct that the term “meta-plasticity” might be confusing to readers. Following the reviewer’s suggestion, we modified the text to replace the use of “meta-plasticity” with “meta-learning” throughout the manuscript.

6. **Pg 10 “Despite this, the alignment angles of FeHebb (Fig. S2) are superior to Fbio’s, ”. The word superior is a bit confusing, since here performance is better for Fbio. Instead, I think you mean: are better aligned with the backprop direction. It is not clear (nor likely true) that backprop is the best direction.**

Thank you for catching this inaccurate phrasing. We changed “superior” to “better aligned” (lines 247-248).

7. **Figure 6 was a bit vague. I wonder if this could be augmented to help better understand the description in the text. I didn’t much follow it. Maybe adding the update equations there might help understand.**

To help clarify the interpretation of Fig. 6, we summarized the update equations in the caption.

Information flow between the forward and backward pathways: (a) Both layers are trained with the rule $\mathcal{F}(\Theta) = \theta_0 \mathcal{F}^0$ via feedback alignment. In this case, information from $B_{2,1}$ is transmitted to $W_{0,1}$ through \mathcal{F}^0 (①) and then propagated forward to $W_{1,2}$ (②). (b) The first layer is updated with the rule $\mathcal{F}(\Theta) = \theta_0 \mathcal{F}^0$ via feedback alignment, while the second layer uses $\mathcal{F}^{eHebb}(\Theta) = \theta_0 \mathcal{F}^0 + \theta_2 \mathcal{F}^2$. Using \mathcal{F}^0 , information from $B_{2,1}$ is communicated to $W_{1,2}$ (①, ③); meanwhile, the presence of \mathcal{F}^2 sets up a new channel to directly communicate information from $B_{2,1}$ to $W_{1,2}$ (②). The blue arrows depict information propagation through the forward and backward paths. The communications between feedback and feedforward pathways are represented with red arrows.

We also added and modified the text to support these changes and further explain the figure (lines 262-265).

Note that the mechanism in \mathcal{F}^0 needs two learning iterations to transmit information from $\mathbf{B}_{2,1}$ to $\mathbf{W}_{1,2}$; information from $\mathbf{W}_{0,1}$ propagates to $\mathbf{W}_{1,2}$ only after \mathbf{y}_1 is computed with the updated $\mathbf{W}_{0,1}$. Meanwhile, \mathcal{F}^2 does this in the same iteration, carrying out expedited learning.

Similarly, we modified the description for the diagrams shown in Fig. S3.

Interactions between feedback and forward pathways using \mathcal{F}^{eHebb} : (a) All layers trained with the rule $\mathcal{F}(\Theta) = \theta_0 \mathcal{F}^0$ via feedback alignment. Information from $\mathbf{B}_{3,2}$ and $\mathbf{B}_{2,1}$ is transmitted to $\mathbf{W}_{0,1}$ through the \mathcal{F}^0 plasticity rule (①), which then passes on to $\mathbf{W}_{1,2}$ and $\mathbf{W}_{2,3}$ (③). Meanwhile, information from $\mathbf{B}_{3,2}$ is transmitted to $\mathbf{W}_{1,2}$ (②), which is then propagated to $\mathbf{W}_{2,3}$ after the forward propagation. (b) $\mathbf{W}_{2,3}$ is updated using $\mathcal{F}^{eHebb}(\Theta) = \theta_0 \mathcal{F}^0 + \theta_2 \mathcal{F}^2$, while $\mathbf{W}_{0,1}$ and $\mathbf{W}_{1,2}$ are trained with the rule $\mathcal{F}(\Theta) = \theta_0 \mathcal{F}^0$ via feedback alignment. Plasticity rule \mathcal{F}^0 transmits information from $\mathbf{B}_{2,1}$ and $\mathbf{B}_{3,2}$ to $\mathbf{W}_{0,1}$ (①) and from $\mathbf{B}_{3,2}$ to $\mathbf{W}_{1,2}$ (②). This information is propagated to their downstream layers after the forward path (④). Concurrently, an additional channel established by \mathcal{F}^2 explicitly propagates the information from $\mathbf{B}_{3,2}$ to $\mathbf{W}_{2,3}$ (③). (c) $\mathbf{W}_{0,1}$ and $\mathbf{W}_{2,3}$ use the plasticity rule $\mathcal{F}(\Theta) = \theta_0 \mathcal{F}^0$ via feedback alignment, and $\mathbf{W}_{1,2}$ utilizes $\mathcal{F}^{eHebb}(\Theta) = \theta_0 \mathcal{F}^0 + \theta_2 \mathcal{F}^2$. \mathcal{F}^0 communicates information from $\mathbf{B}_{2,1}$ and $\mathbf{B}_{3,2}$ to $\mathbf{W}_{0,1}$ (①), which then is propagated to the downstream layers (③). Meanwhile, the \mathcal{F}^0 rule in \mathcal{F}^{eHebb} disseminates information from $\mathbf{B}_{3,2}$ to $\mathbf{W}_{1,2}$, while \mathcal{F}^2 in \mathcal{F}^{eHebb} establishes a direct route to transmit information from $\mathbf{B}_{2,1}$ to $\mathbf{W}_{1,2}$ (②). The ensuing forward propagation from $\mathbf{W}_{1,2}$ to the downstream layers continues as usual. In all graphs, blue arrows represent the propagation of data through the forward or backward path, while the red arrow represents the flow of information from the backward pathway to the forward connections.

8. Table 1 is confusing. What are the α_0 , α_1 and α_2 for? Just for the Hebbian angles, or somehow a ratio between the angles for Hebbian versus random alignment?

We thank the reviewer for catching the omission of these details. α_ℓ represents the alignment angle between the teaching signal e_ℓ and the backprop direction. The subscript ℓ denotes the layer index. We modified the caption for clarity (Table 1).

9. How did you pick the 9 possible plasticity rules? Are they all biologically plausible rules?

Since we did not restrict ourselves to previously studied plasticity rules, there was an unlimited number of potential rules to choose from. Some of the rules were chosen because they match previously studied rules (\mathcal{F}^0 is from backprop, \mathcal{F}^3 is weight decay, \mathcal{F}^9 is Oja's rule). We also included all quadratic combinations of errors and activations (\mathcal{F}^0 - \mathcal{F}^2) except for pure Hebbian plasticity ($\mathbf{y}_\ell \mathbf{y}_{\ell-1}^T$) because it lead to unstable dynamics. We therefore replace pure Hebbian plasticity with Oja's rule, which adds a stabilizing term to pure Hebbian plasticity. We also included some higher order terms to test their viability. The terms are biologically plausible in the sense that they only combine activations, errors, and weights that are local to the modified

synapse. In response to this comment and a comment from Reviewer 2, we added an in depth discussion of the biological plausibility of our plasticity rules to the Discussion (lines 374-409)

While synaptic plasticity in the brain is mediated by a vast array of biophysical processes, the changes to a single synaptic weight largely depend on the activity of its pre-synaptic and post-synaptic neurons and the current weight, a property known as “local” plasticity. For the plasticity rules used in our study (with the exception of Oja’s rule), weight updates depend on activations from a forward pass *and* error signals from a backward pass. Since these quantities were used to update the *forward* projecting weights, this raises the question of whether the plasticity rules are truly local. The answer to this question depends on the biological interpretation of the forward and backward passes.

Under one interpretation, separate populations of neurons encode the forward and backward passes, *i.e.*, the neurons encoding e_ℓ are distinct from those encoding y_ℓ . Under this interpretation, the plasticity rules used in this study are not strictly local.

Under another interpretation, forward activations and backward errors are represented by the same neural populations, *i.e.*, the same neurons encode e_ℓ and y_ℓ . Under this interpretation, all of the plasticity rules used in this study are local. There are several models for how this multiplexing of forward and backward signals could be achieved (see [7] for a review). For example, activations and errors could be represented at separate points in time by the same neurons.

Alternatively, recent work hypothesizes that activations and errors are encoded separately in the basal and apical dendrites of the same cortical pyramidal neurons [8]. Along similar lines, a growing body of work posits that activations and errors are multiplexed by the distinction between bursts and single action potentials, which are communicated separately by synaptic projections onto the soma versus apical dendrites of pyramidal neurons [9, 10, 11]. The dependence of synaptic plasticity on the morphological site of the synaptic contact and on the type of spiking (bursts versus individual spikes) is well established in experiments [12, 13, 14, 15, 16]. Under these models, established biophysical properties of cortical synapses can produce plasticity rules like ours that multiplex forward and backward propagating information to update weights. Networks in [11] rely on weight decay to approximately align forward and backward weights [17], while some networks in [8] rely on random feedback alignment. Hence, our meta-learned plasticity rules could improve learning in those models.

Our meta-learning approach isolated three plasticity terms: a backprop-like rule (\mathcal{F}^0), Oja’s rule [18] (\mathcal{F}^9), and a rule we refer to as eHebb (\mathcal{F}^2). Possible biological implementations of Oja’s rule and the backprop-like rule have been studied in great depth in previous work [8, 7, 19, 11]. The eHebb rule could be implemented in a similar way to the backprop-like rule. For example, under the model in [11], eHebb would change synaptic weights in response to the co-occurrence of pre- and post-synaptic bursts. Plasticity is strongly mediated by firing rates and intracellular calcium [20, 21], both of which are elevated during bursts.

In addition, to describe our choice of plasticity rules, we added the following text to the Meth-

ods where the plasticity rules are introduced (lines 475-483):

The rules above are local in the sense that the updates to the j, k th entry of $\mathbf{W}_{\ell-1, \ell}$ depend only on the k th entry of $e_{\ell-1}$ and $\mathbf{y}_{\ell-1}$, the j th entry of \mathbf{y}_{ℓ} and e_{ℓ} , and the j, k th entry of $\mathbf{W}_{\ell-1, \ell}$. This notion of locality assumes that errors and activations are encoded in the same neurons (see Discussion). Even under this constraint of locality, there is an unlimited number of possible plasticity rules to choose from. To form the list above, we first considered all quadratic combinations of activations and errors except we omitted pure Hebbian plasticity ($\mathbf{y}_{\ell} \mathbf{y}_{\ell-1}^T$) because we found that it leads to unstable network dynamics (a blowup of activations). Instead, we replaced it with Oja’s rule \mathcal{F}^O , which adds a stabilizing term onto pure Hebbian plasticity. Additional terms were added to test the viability of higher order plasticity terms.

10. **The text states that adding Oja’s rule improves performance, but not alignment. But, it seems to me that alignment for Fbio and when just using Oja’s (plus the gradient term) is not *too* different. Is the implication that Fbio has aligned well, but Oja’s rule has not?**

Figure 7c shows the alignment of the teaching signal of \mathcal{F}^{Oja} with the backprop analog. Figure S2c shows the same plot for \mathcal{F}^{bio} . While the alignment angles in \mathcal{F}^{Oja} are, on average, around 75° for all layers, using \mathcal{F}^{bio} , this reduces to as low as 35° for the deeper layers. Thus, in the deeper layers, the teaching signals of \mathcal{F}^{bio} are substantially more aligned with the back-propagated teaching signals. We modified the text to clarify this (lines 292-295):

In fact, alignment angles are only slightly smaller when using Oja’s rule compared to using pure FA, as seen by comparing Fig. 7c to Fig. 3d. This contrasts with alignment angles for \mathcal{F}^{eHebb} and \mathcal{F}^{bio} , which are greatly reduced in deeper layers compared to \mathcal{F}^{Oja} (compare Fig. 7c to Figs. 5c and S2c).

11. **Finally, you did a nice thorough analysis on this dataset. That is enough. But it would also be interesting to see if this behavior of Oja’s rule extends to other datasets, or if we see that other rules are much more effective there. This is not a criticism so much as: your work has intrigued me and I want to know more.**

We thank the reviewer for this excellent suggestion to test Oja’s rule under our learning framework on another dataset. Oja’s rule is effective, at least in part, because of how it reduces correlations within rows of the weight matrices. We expect this property to be independent of the dataset and therefore expected to see benefits of Oja’s rule in other data sets. To test this empirically, we evaluated the performance of the \mathcal{F}^{Oja} in a 10-way classification on the FashionMNIST dataset and compared the results with Feedback Alignment and backprop methods on the same dataset (lines 705-710). Results show substantial improvement in learning through \mathcal{F}^{Oja} in the presence of random feedback connections (Fig. S4).

Performance of the \mathcal{F}^{Oja} on FashionMNIST

Section 2.2.2 examines how Oja’s rule improves learning in the Feedback Alignment model (Fig. 7). In this section, we demonstrate the effectiveness of Oja’s rule

on a different dataset by using the FashionMNIST [22] to train a classifier model. Figure S4 illustrates that introducing the Oja’s rule (Eq. 9) substantially enhances learning across different datasets when the model is trained with random feedback connections.

Figure S4: **Performance of benchmark learning schemes** while training a 5-layer fully-connected classifier network on FashionMNIST dataset [22] for a 10-way classification task. The plot demonstrates accuracy versus the number of training data for Feedback Alignment (FA) [23] and backprop (BP) [24] methods, compared to \mathcal{F}^{Oja} (Eq. 9).

Studying different learning rules in the a *meta-learning* framework requires a proper dataset (as discussed in Section 4.3). Such a dataset should have

- Many classes to sample different tasks.
- Enough samples per class to simulate learning in a lifetime.
- Appropriate for the classifier network, i.e., a fully-connected network.

However, such datasets are lacking in the literature. Thus, exploring other datasets in the present meta-learning framework is subject to the availability of appropriate datasets.

12. **Eq 6 should have $L_{meta}(\Theta)$, so it is clear that the objective function is for Θ .**

We thank the reviewer for the suggestion. We modified the meta-objective function for clarification (Eq. 6).

13. **Figure 2 will also be more useful once you introduce the functional form for \mathcal{F} earlier. This figure was not useful to me until I read 2.2.2.**

We thank the reviewer for this suggestion, which we addressed in response to the reviewer’s earlier comment (Minor-1).

Reviewer 2

Shervani-Tabar and Rosenbaum explore possible learning rules with the aim to boost the performance of a feedforward network on a classification task that relies on random feedback alignment to back propagate errors through the network. The authors propose 10 mathematical terms and use a meta-learning approach to show that three of the explored terms are sufficient to increase the accuracy of the network to similar levels as when it is trained with a backprop algorithm.

The manuscript is well written and the results are interesting, shining a light on how the brain (with biological constraints) might implement algorithms that are similar to backprop. I appreciate the analysis of the individual plasticity terms that the authors find with their meta-learning approach. I also consider it relevant because it explores possible improvements to the random alignment algorithm. It is unclear to me, however, how pre- and post-synaptic errors would be encoded and transmitted to specific synapses so that the meta-learned rules are biologically plausible, as claimed by the authors. I have other specific points which I elaborate below on (major and minor points) which I think are important to be addressed by the authors.

Major

1. It is unclear whether the rules discovered by the author’s meta-learning approach are indeed biologically plausible. Apart from Oja’s rule, which is solely composed of pre- and post-synaptic activities as well as the connection weight, the other two “bio” terms (referred to as “eHebb”) use pre- and post-synaptic errors that are presumably computed by a backward pathway. It seems to me that this makes the rule nonlocal and the concept of biological plausibility may be lost. In this vein, it would be interesting to compare the author’s model to biophysical implementation of backprop in Sacramento *et al.* (NeurIPS 2018). Could the authors elaborate how the learning rule found by their meta-learning approach could be implemented with biological constraints?

We thank the reviewer for catching some implicit assumptions that were made by interpreting our plasticity rules as local. Plasticity rules that involve backward projecting error terms are only local under biological interpretations in which errors and activations are encoded by the same neurons. They are not local if errors and activations are encoded by separate neural populations. There is a rich literature on biological models and interpretations of forward and backward passes. Many inter (including the one in Sacramento *et al.* [8]) use the same neurons to encode forward and backward passes, and would therefore interpret our plasticity rules as local. To better explain the “locality” and biological plausibility of our plasticity rules, and to relate our work to Sacramento *et al.* and other studies, we added the following text to the Discussion (lines 374-409):

While synaptic plasticity in the brain is mediated by a vast array of biophysical processes, the changes to a single synaptic weight largely depend on the activity of its pre-synaptic and post-synaptic neurons and the current weight, a property known as “local” plasticity. For the plasticity rules used in our study (with the exception of Oja’s rule), weight updates depend on activations from a forward pass *and* error signals from a backward pass. Since these quantities were used to update the *forward* projecting weights, this raises the question of whether the plasticity rules

are truly local. The answer to this question depends on the biological interpretation of the forward and backward passes.

Under one interpretation, separate populations of neurons encode the forward and backward passes, *i.e.*, the neurons encoding e_ℓ are distinct from those encoding \mathbf{y}_ℓ . Under this interpretation, the plasticity rules used in this study are not strictly local.

Under another interpretation, forward activations and backward errors are represented by the same neural populations, *i.e.*, the same neurons encode e_ℓ and \mathbf{y}_ℓ . Under this interpretation, all of the plasticity rules used in this study are local. There are several models for how this multiplexing of forward and backward signals could be achieved (see [7] for a review). For example, activations and errors could be represented at separate points in time by the same neurons.

Alternatively, recent work hypothesizes that activations and errors are encoded separately in the basal and apical dendrites of the same cortical pyramidal neurons [8]. Along similar lines, a growing body of work posits that activations and errors are multiplexed by the distinction between bursts and single action potentials, which are communicated separately by synaptic projections onto the soma versus apical dendrites of pyramidal neurons [9, 10, 11]. The dependence of synaptic plasticity on the morphological site of the synaptic contact and on the type of spiking (bursts versus individual spikes) is well established in experiments [12, 13, 14, 15, 16]. Under these models, established biophysical properties of cortical synapses can produce plasticity rules like ours that multiplex forward and backward propagating information to update weights. Networks in [11] rely on weight decay to approximately align forward and backward weights [17], while some networks in [8] rely on random feedback alignment. Hence, our meta-learned plasticity rules could improve learning in those models.

Our meta-learning approach isolated three plasticity terms: a backprop-like rule (\mathcal{F}^0), Oja’s rule [18] (\mathcal{F}^9), and a rule we refer to as eHebb (\mathcal{F}^2). Possible biological implementations of Oja’s rule and the backprop-like rule have been studied in great depth in previous work [8, 7, 19, 11]. The eHebb rule could be implemented in a similar way to the backprop-like rule. For example, under the model in [11], eHebb would change synaptic weights in response to the co-occurrence of pre- and post-synaptic bursts. Plasticity is strongly mediated by firing rates and intracellular calcium [20, 21], both of which are elevated during bursts.

We also added the following text to the Methods section where the plasticity terms are given (lines 475-478):

The rules above are local in the sense that the updates to the j, k th entry of $\mathbf{W}_{\ell-1, \ell}$ depend only on the k th entry of $e_{\ell-1}$ and $\mathbf{y}_{\ell-1}$, the j th entry of \mathbf{y}_ℓ and e_ℓ , and the j, k th entry of $\mathbf{W}_{\ell-1, \ell}$. This notion of locality assumes that errors and activations are encoded in the same neurons (see Discussion).

2. **The meta-learning algorithm (Algorithm 1) does not seem to be entirely novel as described by the authors in the abstract: ”In this study, we develop a novel meta-plasticity approach to discover interpretable, biologically plausible plasticity rules.. ” (see, e.g., Confraveux**

et al. 2020). In my view, what seems to be novel is the set of terms used in the search space of the meta-learning algorithm (Section 4.2). Could the authors elaborate on the precise novelty in the algorithm?

As per the editor’s request and following the journal’s policy, we avoid the word “novel” in the revised version of the manuscript.

Our method solves the weight alignment problem by incorporating meta-parameter sharing, L1 regularization of meta-parameters, online learning, and re-initialized weights. Ours is the first study in which these features have been combined to address weight alignment. Moreover, our analysis of the meta-learned plasticity rules, including Oja’s and Hebbian plasticity between errors, demonstrates how they can overcome the weight alignment problem. These plasticity rules were not previously known to overcome the weight alignment problem.

In contrast, Confraveux *et al.* [25] used meta-learning to recover plasticity rules that were already known to accomplish the task in question, e.g., finding principle components. Moreover, Confraveux *et al.* did not consider error-based learning, which is the focus of this work. We have revised the manuscript to state these points more clearly (lines 113-116).

Previous studies have employed different combinations of elements, such as meta-parameter sharing (as used in [25]) and online learning (as used in [5]). In contrast to previous studies, we integrate all these features to address the weight alignment problem. Our analysis of the meta-learned plasticity rules demonstrates how they overcome the weight alignment challenge.

3. **The authors argue that imposing an L1 penalty on the plasticity coefficients encourages their algorithm to learn a plasticity rule with fewer terms. This is a valid point, however, it should be confirmed with, e.g., comparing it to using L2 (or maybe not using a penalty term if that’s possible). Is it truly a key feature of the author’s approach or would the results be similar with a different norm? I might be missing the point here because it sounds like a technical detail rather than a key feature.**

We thank the reviewer for this suggestion. In order to make a comparison, we have performed experiments on meta-loss with L2 regularization as well as without regularization. The results of these tests can be seen in Fig. S5. Additionally, we have included a section in the supplementary material discussing these results, which can be found on lines 711-727.

Performance of alternative penalization methods

In Sec. 2.2, we proposed using L1 regularization on the meta-loss to decrease redundancy within the update rules. As shown in Fig. 4d, this technique leads to a sparser set of meta-parameters and acts as a model selection method, identifying the most effective plasticity rules.

In Fig. S5, we examine the impact of alternative regularization methods on the meta-learning algorithm by comparing the performance of models with no regularization and L2 regularization. When using no regularization in the meta-learning, the algorithm eliminates update terms negatively impacting the learning. However,

another set of plasticity rules may individually improve the results, but when these rules are considered in a set, other terms may be more beneficial for the optimization process. Nevertheless, the model still includes them in the final meta-optimized learning rule. As seen in Fig. S5a, the model has identified seven plasticity terms, making it impractical to investigate each of these terms individually.

As an alternative, Fig. S5b shows the results of using L2 regularization

$$\mathcal{L}_{meta}(\Theta) = \mathcal{L}(f_{\mathbf{W}}(\mathbf{X}_{query}), \mathbf{Y}_{query}) + \lambda \|\Theta\|_2. \quad (1)$$

Unlike L1 regularization, L2 tends to decrease all parameters but does not return sparse solutions and is unsuitable for feature selection. In other words, even though L2 regularization reduces the values of all parameters, it does not eliminate the redundant or less influential plasticity terms with large meta-parameters from the final solution.

Figure S5: **L1 improves feature selection in the meta-learning model:** Performance of different penalization methods while training a 5-layer fully-connected classifier network on EMNIST digits [26] with online learning. Evolution of meta-parameters for the pool of learning rules defined in section 4.2 using (a) no penalization, (b) L2 penalized meta-loss (Eq. S.3).

4. For the candidate learning terms (Section 4.2), all terms are of increasing complexity, yet "simple" multiplications of different variables apart from the last term, F9, which is Oja's rule. It is not clear to me why a standard Hebb term, $y_l * y_{l-1}^T$, as well as the Oja's penalty term, $y_l * y_{l-1}^T * W_{l-1,l}$, were not used as independent candidate terms, which would make the meta-learning search more independent/robust. Could the authors elaborate on why they have chosen to explicitly use Oja's rule? Additionally, what would happen if Oja's rule was separated into two terms, a purely Hebbian and the penalty term?

We thank the reviewer for catching that we overlooked the explanation for including Oja's rule as a single term instead of each sub-term independently. We found that the standard Hebbian term leads to unstable network dynamics (a blowup of activations after several episodes) when it is included as a term by itself, so we only included it within Oja's rule. To explain this choice, we added the following text to the Methods where the plasticity rules are listed (lines 481-483):

we omitted pure Hebbian plasticity ($\mathbf{y}_l \mathbf{y}_{l-1}^T$) because we found that it leads to unstable network dynamics (a blowup of activations). Instead, we replaced it with Oja's rule \mathcal{F}^9 , which adds a stabilizing term onto pure Hebbian plasticity.

5. The meta-learning algorithm finds 3 terms as relevant to solve the classification task chosen by the authors (as shown in fig. 4), however, one of these terms does not seem to be needed if I interpreted fig. 7 correctly. This raises the question of whether the outer (meta-learning) loop reached a global minimum after 500 episodes (fig. 4), given that the error is similar to backprop with (fig. 4) and without F2 (fig. 7). Could the authors please clarify whether this interpretation of fig. 7 makes sense? I would like to suggest running longer sims (with more episodes) to make sure that a minimum has been reached in the outer loop.

The reviewer expressed a concern about whether rule \mathcal{F}^2 (Hebbian plasticity on errors) is needed to achieve the accuracy achieved by Oja’s rule (\mathcal{F}^0) and \mathcal{F}^9 alone. We would first like to point out that including \mathcal{F}^2 does improve the accuracy obtained by Oja’s rule and \mathcal{F}^9 . This is most visible by comparing the performance with Oja’s rule and \mathcal{F}^9 (Fig. 7a) to the performance with Oja’s, \mathcal{F}^0 , and \mathcal{F}^2 combined (Supplementary Fig. S2a). In making this comparison, we observe an improvement of approximately 5 – 10% when \mathcal{F}^2 is included in the plasticity rule. For the reviewer’s convenience, we have included a comparison of these two plots below.

Figure 1: Comparison between the accuracy of the classifier using the plasticity rules $\mathcal{F}^{Oja}(\Theta) = \theta_0\mathcal{F}^0 + \theta_9\mathcal{F}^9$ (black) and $\mathcal{F}^{bio}(\Theta) = \theta_0\mathcal{F}^0 + \theta_2\mathcal{F}^2 + \theta_9\mathcal{F}^9$ (red).

We appreciate the reviewer’s suggestion to run longer meta-learning experiments. In response, we now run all meta-learning tests for a total of 600 episodes.

Minor

1. The term ”meta-plasticity” is usually used in neuroscience to refer to the ”plasticity of synaptic plasticity” (see, e.g., Abraham and Bear, 1996), which differs in meaning to ”meta-learning”, usually used in machine learning for automatic algorithm learning (well explained by the authors in page 5, line 143). For example, ”meta-plasticity” is used in page 2, line 68; page 3, line 78; page 3, line 83; and other places, where it means (as far as I understand) ”meta-learning”.

We appreciate the reviewer bringing attention to the potential conflict in terminology. We have replaced the term ”meta-plasticity” with ”meta-learning” throughout the manuscript.

2. Page 3, line 83: ”We use meta-plasticity to learn a parameterized plasticity rule based on a combination of biologically motivated rules.” It is not clear to me how these terms (outlined in section 4.2) are biologically motivated (please see point 1).

We changed “biologically motivated rules” to “candidate rules” and we relegate the discussion of the rules’ biological plausibility to the Discussion (see reply to Major-1 above) where it can be addressed in sufficient detail.

3. **Page 3, line 86-91: The first two key features presented by the authors seem to have considerable overlap with Confavreux et al., 2020. If that’s the case, I would like to suggest adding a citation there and clarifying this in the text.**

We appreciate the reviewer’s suggestion. We have revised the text to cite Confavreux *et al.* [25] and clarify their use of meta-parameter sharing. Regarding the re-initialization of weights, they mention that they begin meta-training the network from randomly initialized weights. However, it is not specified in their paper if the network weights are reinitialized to a random value after each meta-iteration, so we did not mention whether they use this approach. But we did rephrase our description of this approach to avoid claiming that it is novel in our work.

4. **Page 3, line 92: It is not clear to me how relevant this feature is for the results found by the authors. Would the method not work using L2? (please see point 3).**

We appreciate the reviewer for bringing this point to our attention. We have addressed this issue in response to a previous comment (Major 3).

5. **Page 3, line 95: Previous work on meta-learning plasticity rules (e.g., Confavreux et al., 2020; Lindsey and Litwin-Kumar, 2020) have always considered “online rules”. Could the authors elaborate on which would be the other possible type of implementation apart from the online learning in their setting? Why is this a key feature of the authors’ approach?**

We would first like to point out that the term “online learning” has various interpretations in the machine learning literature. In our manuscript, we adopt the definition by Goodfellow *et al.* [6], which refers to the training using a batch size of one for a single epoch. In the context of meta-learning, this means updating the inner loop one data point at a time with K gradient descent updates using a set of K samples from task \mathcal{T}_i . Finally, the meta-loss is computed on the query set of task \mathcal{T}_i based on the K th weight update. It is noteworthy that during each episode, only one task is observed. This approach, sometimes called Online-aware Meta-Learning [27], helps to avoid catastrophic interference.

The reviewer correctly notes that Lindsey *et al.* [5] use online learning in their model. We have revised our text to acknowledge this (line 114). In contrast, Confavreux *et al.* [25] use batch learning within their inner loop; i.e., for each task \mathcal{T}_n , the model is updated for E epochs, each with B gradient descent updates, where B is the number of batches per epoch. As stated in the caption of Figure 1 in their paper, the networks were trained using a batch size of 200 for the weight updates. Subsequently, the meta-loss is computed across multiple tasks. During each episode, N networks are trained over N unique tasks \mathcal{T}_n for E epochs, each with B batches per epoch. Finally, the meta-loss is determined as the average loss of these N networks on the query set of their task \mathcal{T}_n .

In the present work, our goal is to improve the feedback alignment technique by refining the learning rule, and online learning is particularly relevant in this setting. A challenge encountered by the feedback alignment technique and other asymmetric feedback methods is that utilizing small batch sizes can lead to a decline in performance. Therefore, it is crucial for the

plasticity rule obtained through meta-learning to effectively train the model while learning from an online stream of data. To emphasize the pertinence of online learning for the weight alignment problem addressed in our study, we added a comment about this when we first introduced our use of online learning (line 112).

- 6. Page 4: The authors explain well the backprop algorithm, but I find it confusing to have Eq. 4 defining the matrix B as the transpose of W , given that B is also used for the random feedback alignment algorithm. The authors later in the page define superscripts "FA" and "BP" for error computed with random feedback alignment and backprop, respectively, but these superscripts are not used in, e.g., page 6, when they define the meta-learning algorithm. I would like to suggest to the authors to always use a superscript in both error and matrix B (starting in Eq. 3) to clearly state when they are referring to the transpose of W or a fixed matrix.**

We appreciate the reviewer's feedback on the potential confusion regarding the feedback connections in backprop. In response, we have updated the manuscript (lines 127-137) to clearly distinguish B using superscripts FA and BP when strictly referring to each method. However, in several instances, we are not strictly referring to either method. For example, Algorithm 1 is used for both the Feedback Alignment and backprop methods, as shown in Fig. 3. Therefore, we drop the superscript when we do not specifically refer to either case.

- 7. Page 4, line 131: "Alternative approaches have proposed a direct feedback pathway... but we find that this also does not achieve backprop performance" Could the authors elaborate? I think the manuscript would be more clear with a longer explanation of these findings here.**

We appreciate the reviewer's suggestion. We have revised the text to provide more information about the Direct Feedback Alignment method and the performance of this model (lines 151-156).

An alternative approach to using feedback connections that link consecutive layers is to create direct backward pathways [28]. This change allows errors to be transmitted directly from the output layer to the upstream layers. This modification leads to improved performance compared to the feedback alignment method, speeding up the learning process and improving accuracy. However, it still falls short of the performance level of backpropagation (see Supplementary Fig. S1).

For further clarification, a comprehensive description of the DFA has been added to the Supplementary Material (lines 649-661).

Figure 1 illustrates that the Feedback Alignment model [23] is less effective than the backprop model when training deep networks with a continuous data stream. To be more precise, the backprop model begins learning immediately at the start of training, while the Feedback Alignment model takes around 2000 training data points before it starts to learn. Additionally, the rate of learning for the Feedback Alignment model is slower.

In an attempt to improve the Feedback Alignment model's performance, the Direct Feedback Alignment (DFA) method [28] proposed altering the backward connections to directly transmit errors from the output layer y_L to the upstream layers

\mathbf{y}_ℓ . The modulating signals in this modified model are calculated as

$$\mathbf{e}_\ell = \mathbf{B}_{L,\ell} \mathbf{e}_L \odot \sigma'(z_\ell),$$

with

$$\mathbf{e}_L = \frac{\partial \mathcal{L}}{\partial z_L}.$$

In this formulation, $\mathbf{B}_{L,\ell} \in \mathbb{R}^{\dim(\mathbf{y}_\ell) \times \dim(\mathbf{y}_L)}$, where $\dim(\mathbf{y}_\ell)$ represents the dimensionality of the activation \mathbf{y}_ℓ .

As shown in Fig. S1, incorporating direct feedback connections to the Feedback Alignment method speeds up learning, and the model’s accuracy improves after 1000 training data points. However, even with this modification, the network’s performance is still lower than that of the backprop model. Figure S1 further compares the DFA model with the Feedback Alignment model trained with the \mathcal{F}^{bio} plasticity rule (Eq. 7) and shows that the improved plasticity rule outperforms the DFA model.

8. Page 5, fig 1a: How is the matrix B generated for FA? The "speed" by which it reaches high accuracy is always the same for different initialisations of B?

We randomly sampled forward and backward connections from a uniform distribution, utilizing the Xavier initialization method. We have revised the text to clarify this (lines 465-468).

In the fixed feedback pathway problem, the weights and feedback connections are initially set to random values that differ from each other. Both symmetric and fixed feedback models utilize the Xavier method [29] to re-initialize forward and backward connections at the start of each meta-learning episode.

To address the reviewer’s question, we have modified Fig. 1 to demonstrate the model performance across different trials, each starting from a different initial value. Furthermore, we assessed the model’s performance with different initialization methods, using the normal distribution for backward connections and uniform distribution for forward connections (lines 728-739). Our results, presented in Fig. S6, demonstrate the model’s capability to learn effectively using the proposed plasticity rule under various initial values and initialization methods.

Performance of alternative backward initialization

As mentioned in Sec. 4.1, the Xavier initialization method was used to randomly sample forward and backward connections from a uniform distribution

$$\mathbf{B}_{\ell+1,\ell}, \mathbf{W}_{\ell,\ell+1} \sim \mathcal{U} \left(-\sqrt{\frac{6}{\dim(\mathbf{y}_\ell) + \dim(\mathbf{y}_{\ell+1})}}, \sqrt{\frac{6}{\dim(\mathbf{y}_\ell) + \dim(\mathbf{y}_{\ell+1})}} \right)$$

throughout the study, where $\dim(\mathbf{y}_\ell)$ is the dimension of the activation \mathbf{y}_ℓ . Nevertheless, the findings presented in this work do not depend on the initialization method of the backward connections.

To illustrate this, we conducted an experiment where we employed the normal Xavier initialization method

$$B_{\ell+1,\ell} \sim \mathcal{N}\left(0, \frac{2}{\dim(\mathbf{y}_\ell) + \dim(\mathbf{y}_{\ell+1})}\right)$$

to sample initial values for the backward connections. The forward connections were initialized using a uniform distribution as before (Eq. S.4). Figure S6 shows that the proposed \mathcal{F}^{bio} plasticity rule can successfully train the model using different methods for initializing the backward connections.

Figure S6: \mathcal{F}^{bio} trains effectively under different initialization of the feedback: Accuracy of a 5-layer classifier network trained on MNIST dataset [30] to perform a 10-way classification task using Feedback Alignment (FA) [23] compared to the proposed \mathcal{F}^{bio} plasticity rule (bio) outlined in Eq. 7. The backward connections were initialized in both tests using the normal Xavier initialization method (Eq. S.5).

9. **Page 7, fig 2:** In the schematics (middle of figure), it looks like the activation variables, y_i and y_j , are computed separately from the errors, e_i and e_j . The implementation in a simulation is straightforward, but the biological implementation is not (as intuitively seen in the schematics). Could the authors elaborate on how this type of plasticity would be implemented in a biological setting? I would like to suggest adding such an implementational hypothesis to this figure.

Thank you for catching this oversight in explaining the interpretation of our model. To clarify this point, we added the following sentence to the caption in Fig. 2, which points to the Discussion section where this issue is addressed in more detail (see response to Major-1 above):

Such terms are consistent with local plasticity if y_i and e_i are encoded by the same neuron (see Discussion).

10. **Page 7, line 180:** I would like to suggest explicitly describing panels a to c of figure 3 in the main text to help the readers.

We thank the reviewer for this suggestion. As suggested, we have modified the text to include a description of panels a to c in Fig. 3 in the main text (lines 215-216).

Figures 3a - 3c compare the performance of the two plasticity rules over 600 episodes. First, the reinitialized models f_W are trained at each episode using an online stream

of $M \times K = 250$ data points. Then, the meta-accuracy and meta-loss are evaluated with the query data. Tracing the evolution of the plasticity coefficients in Fig. 3c shows that the meta-learning model converges after ~ 100 episodes. After convergence, the model trained with feedback alignment is, on average, about 25% accurate in its predictions, whereas the model backpropagated via symmetric feedbacks reaches an approximate accuracy of about 70% (Fig. 3a). In addition, the backpropagated model reaches considerably lower loss values as shown in Fig. 3b.

11. **Page 9, (no line numbers on this page): I would like to suggest explicitly describing panels a to c of figure 4 in the main text to help the readers.**

We appreciate the reviewer's suggestion. Per the suggestion, we have revised the text to include a description of panels a, b, and c of Fig. 4 in the main text (page 10).

As seen in Fig. 4a, the model's accuracy initially resembles that of the FA model, but as the meta-optimization continues, the accuracy improves, starting around 10 episodes. By about 300 meta-iterations, the accuracy approaches that of the BP model. This trend is also echoed in Fig. 4b, where the loss initially follows that of the FA learning model but then declines and eventually becomes similar to that of the BP method. In Fig. 4c, it is demonstrated that the alignment angles of the teaching signals with their BP counterparts are improved compared to the FA model, seen in Fig. 3d.

12. **Page 9, fig 4d: It's not clear from the figure whether the system (outer loop) reached a global minimum after 500 meta-learning iterations (episodes). Please see point 5.**

We appreciate the reviewer's suggestion. We have previously addressed this in our response to the previous comment (Major 5).

13. **Page 15, line 295: "The proposed plasticity rules draw upon biological learning rules, including Hebbian plasticity and Oja's rule, ..." To the best of my knowledge, Oja's rule is a type of Hebbian plasticity rule. I would like to suggest clarifying it in the Discussion section.**

We thank the reviewer for this suggestion. We have revised the text to make the distinction clear (lines 338-341).

Our work accelerates the learning process by enhancing the rules that govern neural plasticity while transmitting teaching signals through fixed connections. Our proposed rules for plasticity are based on biologically motivated learning principles, like Oja's rule, or have been inspired by them, such as the error-based Hebbian rule.

14. **Page 15, line 296: Continuing last point; "... to combine information locally from their immediate neighbors, ..." It's not clear to me what the authors mean by "neighbors" in this sentence. The same term is used later in page 16, line 307, without being clear to me either what the term means.**

We intended this language to express the locality of the plasticity rules, but we agree with the reviewer that it was unclear. Given that the locality of our plasticity rules is also more nuanced

(as discussed above), we removed all mentions of “neighbors” and leave the discussion of the locality of the plasticity rules for the Discussion (see response to Major-1 above).

15. **Page 16, line 306: ”The first one uses a Hebbian-style learning term ...” In the previous page the authors refer to it as ”Hebbian plasticity” rather than ”Hebbian-style learning”. I find these terms a bit misleading because, as far as I know, Oja’s rule is an implementation of Hebbian learning and it could be referred to as Hebbian-style learning, however, the authors state that a learning rule with error terms is Hebbian. I would like to suggest clarifying these statements.**

We appreciate the reviewer’s suggestion. We have modified the text to make the distinction clear and easy to understand for the reader (lines 358-361).

The first, an error-based Hebbian rule, combines the errors of pre- and postsynaptic layers to update forward projecting weights. The second rule, known as Oja’s rule, combines pre- and post-synaptic activations with the connection’s current state to update weights.

16. **Page 16, line 335: ”... interpretable learning rules satisfying biological constraints ...” These constraints are not entirely clear to me. The next sentence states that ”... we only considered plasticity in the forward connections...” However, this is, in my view, not a strong biological constraint, such as, e.g., plasticity being local. The manuscript’s clarity would benefit from an introduction of what biological constraints are being imposed to arrive at the terms introduced in section 4.2.**

We thank the reviewer for catching the unclear wording in these sentences. In this paragraph, we meant to point out that our approach could be applied to some different problems than the one we studied. We rephrased the sentences in question as follows (lines 422-428):

We used meta-learning to find plasticity rules that can learn effectively under the biologically relevant setting where forward and backward weights are not explicitly aligned. But our meta-learning technique can be applied more broadly to identify plasticity rules that overcome other biological constraints in various contexts and models. For instance, our study only focused on plasticity in forward connections; however, backward projections in the brain can also exhibit plasticity. Our meta-learning approach can be extended to discover plasticity rules for backward connections in such settings.

17. **Page 16, line 374: ”We focused on metal-learning biologically motivated plasticity rules, ...” Similar to the point above, the manuscript’s clarity would benefit from a more detailed explanation of how each term (or rule) follows a specific biological constraint.**

We changed “biologically motivated” to “biologically plausible” because it more precisely describes the pool of rules we used. An explanation for the biological plausibility of rules was addressed in previous comments.

18. **Page 26, line 529: The ”big” parentheses to the right of ”i,k” and ”j,k”” should be moved to the right so that the sum over k and k’ also takes into account the error terms.**

Thank you for catching this error, which we have corrected.

Reviewer 3

This paper is interesting, well-written, and makes a valuable contribution. I think it should be published once the issues I raise are addressed (either by explaining why they should not be addressed, or, more likely, by modifying the manuscript to address them).

Major

1. I only have one major concern. Throughout the paper in every result there is no discussion of between-run variation. To be scientifically sound, one should do the same experiment multiple times with different random factors (e.g. different initial weights) and then have plots that show intra- and inter-treatment variation. Was each treatment only run once? If so, how can we conclude what is noise vs. signal? Similarly, there should be statistics performed (and proper ones; meaning not assuming normally distributed data unless you first do the proper tests to confirm normality) to show whether the different treatments are statistically significantly different. I recommend redoing the experiments and plotting/reporting bootstrapped confidence intervals, doing statistical tests (e.g. Mann-Whitney tests), etc.

We thank the reviewer for this excellent suggestion. We have repeated all the tests in the manuscript and represent figures with the mean and bootstrap confidence interval. In addition, we have added details of this in the Methods section (lines 471-473).

All plots depict the mean outcome over 20 trials, each with different initial weights and feedback matrices. The shaded region in the loss, accuracy, and meta-parameters plots illustrates the 98% confidence interval, determined through bootstrapping with 500 samples.

To address the reviewer's comment on intra-treatment variations, we have performed the Mann-Whitney test on all the meta-optimization experiments to assess the statistical significance of the accuracy improvements the modified plasticity rules achieved. The results of this analysis (Fig. S7) and the related discussion have been included in the supplementary material (lines 740-752).

Inter-treatment variation

Throughout the paper, we examine the variations within each plasticity rule by calculating the confidence intervals. To determine if the improvements in accuracy are statistically significant, we use the Mann-Whitney U test to compare two sets of data: the accuracy of trials using the FA method and the modified plasticity rule. Samples are taken at the end of each episode and represent the accuracy of the model trained with different initial weights and feedback connection values. We chose the Mann-Whitney U test over the t-test as it does not assume a Gaussian distribution within the groups.

We begin by hypothesizing that the FA method trial samples show lower accuracy than that of the modified plasticity rule. We utilize 20 samples from each

group. The results, illustrated in Fig. S7, indicate that the p-value falls below 5% within fewer than 100 episodes in every example. Our findings indicate strong evidence against the null hypothesis, providing statistical support for the performance gain using the proposed plasticity rules.

Figure S7: The performance gain obtained with the modified plasticity rules is statistically significant: The p-value of the one-sided Mann-Whitney test over 600 meta-optimization episodes, comparing samples from trials using the FA method to those using (a) \mathcal{F}^{eHebb} , (b) \mathcal{F}^{Oja} , (c) \mathcal{F}^{bio} , and (d) \mathcal{F}^{pool} plasticity rules.

Minor

1. **Lines 70-71. Do these papers not have controls that only optimize the initial weights, but without plasticity rules? If so, wouldn't the performance gap between the two tell you the affect/benefit of the learned plasticity rules?**

The reviewer references a study [5] in which the plasticity parameters, feedback connections, and initial forward weights are meta-learned. However, they do not offer any mechanisms to alter these settings. This lack of control makes it impossible to determine the individual impact of each factor.

2. **261-268: It is not clear in these lines how much of the phenomena you describe were known ahead of time (and, if so, which citations support them; there seems to be a lack of sufficient citations given the claims being summarized) vs. new claims/phenomena you are making (in which case supporting data/descriptions seems missing). I think most of these claims were previously known, meaning what is needed is clearer writing and**

more citations. On line 266, citation [26] is at an unconventional location in the sentence, making it difficult for me to figure out what claim that citation is supposed to support.

We thank the reviewer for highlighting a potential source of confusion. As a result, we have made modifications to the text and citations to ensure clarity regarding references to prior works (lines 296-307).

Inspecting Fig. 7 suggests that rather than helping to align the modulating signals, Oja’s rule helps by entirely circumventing the backward path. Oja’s rule implements a Hebbian learning rule subjected to an orthonormality constraint on the weights [18]. In Eq. 9, $\mathbf{y}_{\ell-1}$ and \mathbf{y}_ℓ denote post-nonlinearity activations (as stated in Eq. 2), resulting in the \mathcal{F}^9 plasticity rule to implement a non-linear version of Oja’s rule. When trained iteratively, this non-linear variation implements a recursive non-linear algorithm for Principal Component Analysis [31, 32]. Previous studies on the convergence of Oja’s rule have shown that for a compression layer, where $\dim(\mathbf{y}_{\ell-1}) > \dim(\mathbf{y}_\ell)$, rows of the weight matrix $(\mathbf{W}_{\ell-1,\ell})_1, \dots, (\mathbf{W}_{\ell-1,\ell})_{\dim(\mathbf{y}_\ell)}$ will tend to a rotated basis in the $\dim(\mathbf{y}_\ell)$ -dimensional subspace spanned by the principal directions of the input $\mathbf{y}_{\ell-1}$ [33].

We demonstrate that incorporating Oja’s rule into Feedback Alignment improves feature map extraction in the forward path through unsupervised learning, despite \mathcal{F}^{Oja} not recursively applying pure Oja’s rule.

- 3. Lines 277- 279: Confusing and seemingly contradictory. How is it both true that it is not appropriate to train the last layer this way, but that you did train it that way without any performance degradation (at least, I think that is what you are saying). Overall these lines are unclear. Can you explain more clearly what exactly you did?**

We thank the reviewer for pointing out the unclear phrasing in the manuscript. The initial layers in a classifier network serve as feature extractors, whereas the final layer acts as a predictor, converting the feature representations from the prior layers into the target category for the input data point. Enhancing orthonormality benefits the feature extraction capability of the early layers. However, as the final layer is not a feature extractor, it is not expected to improve the final layer’s performance. Nonetheless, our findings showed no adverse effects when applying the rule to all layers; hence, we chose to use the same plasticity rule to update all layers for consistency. We have modified the manuscript to clarify this remark (lines 317-323).

The architecture of a classifier network includes initial layers that act as feature extractors, creating hidden representations for the final layer. This last layer, dubbed predictor, maps the hidden feature representations to the target class for the given input image. To improve the classifier’s performance, a plasticity rule that enhances feature extraction in the earlier layers is beneficial. However, this rule has no grounds to positively impact the predictor layer’s performance. Despite this, for comprehensiveness, we also applied the plasticity rule \mathcal{F}^{Oja} to the final layer and found no detrimental effect on the model’s performance.

References

- [1] K. Gu, S. Greydanus, L. Metz, N. Maheswaranathan, and J. Sohl-Dickstein. Meta-learning biologically plausible semi-supervised update rules. *bioRxiv*, 2019.
- [2] M. Sandler, M. Vladymyrov, A. Zhmoginov, N. Miller, T. Madams, A. Jackson, and B. A. Y. Arcas. Meta-learning bidirectional update rules. In *International Conference on Machine Learning*, pages 9288–9300. PMLR, 2021.
- [3] M. Andrychowicz, M. Denil, S. Gomez, M. W. Hoffman, D. Pfau, T. Schaul, B. Shillingford, and N. De Freitas. Learning to learn by gradient descent by gradient descent. *Advances in neural information processing systems*, 29, 2016.
- [4] L. Metz, N. Maheswaranathan, B. Cheung, and J. Sohl-Dickstein. Meta-learning update rules for unsupervised representation learning. *arXiv preprint arXiv:1804.00222*, 2018.
- [5] J. Lindsey and A. Litwin-Kumar. Learning to learn with feedback and local plasticity. *Advances in Neural Information Processing Systems*, 33:21213–21223, 2020.
- [6] I. Goodfellow, Y. Bengio, and A. Courville. *Deep learning*. 2016.
- [7] J. C. Whittington and R. Bogacz. Theories of error back-propagation in the brain. *Trends in cognitive sciences*, 23(3):235–250, 2019.
- [8] J. Sacramento, R. Ponte Costa, Y. Bengio, and W. Senn. Dendritic cortical microcircuits approximate the backpropagation algorithm. *Advances in neural information processing systems*, 31, 2018.
- [9] K. P. Körding and P. König. Supervised and unsupervised learning with two sites of synaptic integration. *Journal of computational neuroscience*, 11:207–215, 2001.
- [10] R. Naud and H. Sprekeler. Sparse bursts optimize information transmission in a multiplexed neural code. *Proceedings of the National Academy of Sciences*, 115(27):E6329–E6338, 2018.
- [11] A. Payeur, J. Guerguiev, F. Zenke, B. A. Richards, and R. Naud. Burst-dependent synaptic plasticity can coordinate learning in hierarchical circuits. *Nature neuroscience*, 24(7):1010–1019, 2021.
- [12] O. Paulsen and T. J. Sejnowski. Natural patterns of activity and long-term synaptic plasticity. *Current opinion in neurobiology*, 10(2):172–180, 2000.
- [13] J. J. Letzkus, B. M. Kampa, and G. J. Stuart. Learning rules for spike timing-dependent plasticity depend on dendritic synapse location. *Journal of Neuroscience*, 26(41):10420–10429, 2006.
- [14] B. M. Kampa, J. J. Letzkus, and G. J. Stuart. Requirement of dendritic calcium spikes for induction of spike-timing-dependent synaptic plasticity. *The Journal of physiology*, 574(1):283–290, 2006.
- [15] T. Nevian and B. Sakmann. Spine ca^{2+} signaling in spike-timing-dependent plasticity. *Journal of Neuroscience*, 26(43):11001–11013, 2006.

- [16] R. C. Froemke, I. A. Tsay, M. Raad, J. D. Long, and Y. Dan. Contribution of individual spikes in burst-induced long-term synaptic modification. *Journal of neurophysiology*, 2006.
- [17] M. Akrouf, C. Wilson, P. Humphreys, T. Lillicrap, and D. B. Tweed. Deep learning without weight transport. *Advances in neural information processing systems*, 32, 2019.
- [18] E. Oja. Simplified neuron model as a principal component analyzer. *Journal of mathematical biology*, 15(3):267–273, 1982.
- [19] T. P. Lillicrap, A. Santoro, L. Marris, C. J. Akerman, and G. Hinton. Backpropagation and the brain. *Nature Reviews Neuroscience*, 21(6):335–346, 2020.
- [20] M. Graupner and N. Brunel. Calcium-based plasticity model explains sensitivity of synaptic changes to spike pattern, rate, and dendritic location. *Proceedings of the National Academy of Sciences*, 109(10):3991–3996, 2012.
- [21] M. Graupner, P. Wallisch, and S. Oostjic. Natural firing patterns imply low sensitivity of synaptic plasticity to spike timing compared with firing rate. *Journal of Neuroscience*, 36(44):11238–11258, 2016.
- [22] H. Xiao, K. Rasul, and R. Vollgraf. Fashion-mnist: a novel image dataset for benchmarking machine learning algorithms. *arXiv preprint arXiv:1708.07747*, 2017.
- [23] T. P. Lillicrap, D. Cownden, D. B. Tweed, and C. J. Akerman. Random synaptic feedback weights support error backpropagation for deep learning. *Nature communications*, 7(1):1–10, 2016.
- [24] D. E. Rumelhart, G. E. Hinton, and R. J. Williams. Learning representations by back-propagating errors. *nature*, 323(6088):533–536, 1986.
- [25] B. Confavreux, F. Zenke, E. Agnes, T. Lillicrap, and T. Vogels. A meta-learning approach to (re) discover plasticity rules that carve a desired function into a neural network. *Advances in Neural Information Processing Systems*, 33:16398–16408, 2020.
- [26] G. Cohen, S. Afshar, J. Tapson, and A. Van Schaik. Emnist: Extending mnist to handwritten letters. In *2017 international joint conference on neural networks (IJCNN)*, pages 2921–2926. IEEE, 2017.
- [27] K. Javed and M. White. Meta-learning representations for continual learning. *Advances in Neural Information Processing Systems*, 32, 2019.
- [28] A. Nøkland. Direct feedback alignment provides learning in deep neural networks. *Advances in neural information processing systems*, 29, 2016.
- [29] X. Glorot and Y. Bengio. Understanding the difficulty of training deep feedforward neural networks. In *Proceedings of the thirteenth international conference on artificial intelligence and statistics*, pages 249–256. JMLR Workshop and Conference Proceedings, 2010.
- [30] Y. LeCun, L. Bottou, Y. Bengio, and P. Haffner. Gradient-based learning applied to document recognition. *Proceedings of the IEEE*, 86(11):2278–2324, 1998.

- [31] E. Oja. Data compression, feature extraction, and autoassociation in feedforward neural networks. *Artificial neural networks*, 1991.
- [32] E. Oja. Principal components, minor components, and linear neural networks. *Neural networks*, 5(6):927–935, 1992.
- [33] R. J. Williams. *Feature discovery through error-correction learning*, volume 8501. Institute for Cognitive Science, University of California, San Diego, 1985.

REVIEWERS' COMMENTS

Reviewer #1 (Remarks to the Author):

The authors have largely addressed all of my concerns. The paper is clearly written, the idea simple but very thoroughly explored with clear insights. The methodology is sound and the work is now much better placed with respect to the literature.

Two minor comments:

1. Testing no regularization and l2 was a good idea. However, these plots do not let me see the actual magnitudes of the coefficients. Could you also include the coefficients themselves at the end of learning, in a table, for no regularization, l1 and l2?

2. A reviewer nicely suggested to show behavior across multiple trials. You added this: "All plots depict the mean outcome over 20 trials, each with different initial weights and feedback matrices. The shaded region in the loss, accuracy, and meta-parameters plots illustrates the 98% confidence interval, determined through bootstrapping with 500 samples" What are you bootstrapping, exactly? For bootstrap CIs, I would have expected bootstrapping over the 20 trials themselves.

Reviewer #2 (Remarks to the Author):

The authors have done an excellent job in their replies to my comments! I recommend the publication of the manuscript after consideration of a few minor points (mostly small details) described below. I would like to emphasise that most of them are suggestions to improve the clarity of the manuscript, i.e., for the authors to implement if they consider them to be useful (they don't impact my assessment of the manuscript). The order of the points below are according to the PDF, they do not reflect importance or relevance.

1. x-labels of plots: In Figure 1 and some of the supplementary figures, the x-axis represents episodes of the inner loop (online learning), while in Figures 3, 4, 5, 6, 8 and some supplementary figures, the x-axis represents episodes of the outer loop (meta learning). I would like to suggest changing the labels of the x-axis of each plot from "Episodes" to different words/phrases when the x-axis reflects episodes of the

inner or the outer loop of the algorithm (e.g., "online learning episodes" in one case and "meta learning episodes" in the other).

2. Figure 1b: I would like to suggest adding a bit more details about alpha in the caption of this figure, something similar to the caption of Figure 3d. Additionally, some of the captions (including Figure 1) are very short, and it may be helpful to the reader if more information is included, even if it's the same information from other figure captions. I would like to suggest either repeating the definition of, e.g., alpha (and other relevant information), or referring to the caption where the explanation is more detailed, e.g., "similar to Figure Xa" or "defined in Figure Xa".

3. Page 7, line 193: I appreciate the authors' effort to confirm that the L1 penalty term indeed selects a sparser set of terms in their algorithm compared to L2 or without a penalty term. Based on this confirmation, I would like to suggest to the authors to briefly mention this confirmation in this sentence (page 7, line 193) or to add a brief sentence after this one, pointing the reader to the supplementary material.

4. Page 15, line 315: "These findings indicate that introducing Oja's rule alone can help with the problem of slow learning caused by random feedback connections." It might be worth including a reference to Figure S4 here.

5. Page 15, line 324: "... F^{OJA} provides embeddings that facilitate more effective learning in the last layer." This sentence might be understood as being a bit contradictory with the new text added just above it. It's not entirely clear from this sentence if Oja's rule facilitates learning in the last layer because the last layer follows Oja's rule or because the initial layers follow Oja's rule. In other words, is Oja's rule important when acting in the last layer or it facilitates the learning in the last layer because it changed the activity (or representation) in previous (initial) layers? From the results (and the new added text) it is clear that it is the latter (Oja's rule is important in initial layers, not the last one). The main problem here might be my interpretation of the word "embeddings" and the term "learning in the last layer" in this sentence, so I would like to suggest rephrasing it, being more specific (with details).

6. Page 30, Equation below line 700: There are two "right" parentheses ")" misplaced in the first line of the equation, not the second line. Latex code: $\sum_k (B_{\{l+1,l\}})_{\{i,k\}}(e_{\{l+1\}})_k$ should be $\sum_k (B_{\{l+1,l\}})_{\{i,k\}}(e_{\{l+1\}})_k$. Same for the second term. This way all terms with index k are inside the parentheses where the sum over k is placed.

Reviewer #3 (Remarks to the Author):

The reviewers have addressed my concerns. The one minor request I have is that I think it is appropriate to report medians (not means) plus CIs since you are not making the assumption of normality. I recommend changing to medians (or consulting a statistician to make sure it is ok if you think it is appropriate to stick with means).

Response to reviewers

We thank the reviewers and editor for their helpful comments that have improved the manuscript. Below, we have included a response to each of the reviewers' comments. Reviewer's comments are in **bold text**, our responses are in regular text, and text added to the manuscript is in **red**.

Reviewer 1

The authors have largely addressed all of my concerns. The paper is clearly written, the idea simple but very thoroughly explored with clear insights. The methodology is sound and the work is now much better placed with respect to the literature.

Two minor comments:

- 1. Testing no regularization and l2 was a good idea. However, these plots do not let me see the actual magnitudes of the coefficients. Could you also include the coefficients themselves at the end of learning, in a table, for no regularization, l1 and l2?**

We thank the reviewer for this suggestion. We have modified the manuscript to include a table reporting meta-parameter values after 600 episodes for two different regularization methods (L1 and L2) and with no regularization.

- 2. A reviewer nicely suggested to show behavior across multiple trials. You added this: "All plots depict the mean outcome over 20 trials, each with different initial weights and feedback matrices. The shaded region in the loss, accuracy, and meta-parameters plots illustrates the 98% confidence interval, determined through bootstrapping with 500 samples" What are you bootstrapping, exactly? For bootstrap CIs, I would have expected bootstrapping over the 20 trials themselves.**

We thank the reviewer for pointing out the potentially confusing phrasing in the manuscript. The reviewer is correct that bootstrapping is performed over the trials. In the experiments, we generated 500 bootstrapped samples by sampling 20 observations with resample from the set of 20 trials. We modified the manuscript to clarify this:

The shaded region in the loss, accuracy, and meta-parameters plots illustrates the 98% confidence interval, determined through bootstrapping across trials with 500 bootstrapped samples.

Reviewer 2

The authors have done an excellent job in their replies to my comments! I recommend the publication of the manuscript after consideration of a few minor points (mostly small details) described below. I would like to emphasise that most of them are suggestions to improve the clarity of the manuscript, i.e., for the authors to implement if they consider them to be useful (they don't impact my assessment of the manuscript). The order of the points below are according to the PDF, they do not reflect importance or relevance.

1. **x-labels of plots:** In Figure 1 and some of the supplementary figures, the x-axis represents episodes of the inner loop (online learning), while in Figures 3, 4, 5, 6, 8 and some supplementary figures, the x-axis represents episodes of the outer loop (meta learning). I would like to suggest changing the labels of the x-axis of each plot from "Episodes" to different words/phrases when the x-axis reflects episodes of the inner or the outer loop of the algorithm (e.g., "online learning episodes" in one case and "meta learning episodes" in the other).

We thank the reviewer for pointing out this error. We have changed the x-axis labels in online experiment to "No. training data".

2. **Figure 1b:** I would like to suggest adding a bit more details about alpha in the caption of this figure, something similar to the caption of Figure 3d. Additionally, some of the captions (including Figure 1) are very short, and it may be helpful to the reader if more information is included, even if it's the same information from other figure captions. I would like to suggest either repeating the definition of, e.g., alpha (and other relevant information), or referring to the caption where the explanation is more detailed, e.g., "similar to Figure Xa" or "defined in Figure Xa".

We thank the reviewer for this excellent suggestion. We have modified the caption in Fig. 1 to include information on the alignment angle (α). We have further modified all figures in the manuscript to include more details on the experiments and their results.

3. **Page 7, line 193:** I appreciate the authors' effort to confirm that the L1 penalty term indeed selects a sparser set of terms in their algorithm compared to L2 or without a penalty term. Based on this confirmation, I would like to suggest to the authors to briefly mention this confirmation in this sentence (page 7, line 193) or to add a brief sentence after this one, pointing the reader to the supplementary material.

We thank the reviewer for the suggestion. Based on this suggestion, we have revised the text to direct the reader to the supplementary notes.

4. **Page 15, line 315:** "These findings indicate that introducing Oja's rule alone can help with the problem of slow learning caused by random feedback connections." It might be worth including a reference to Figure S4 here.

We thank the reviewer for the suggestion. Per reviewers suggestion, we have modified the text to point the reader to the supplementary Fig. S4.

5. **Page 15, line 324:** "... F^{OJA} provides embeddings that facilitate more effective learning in the last layer." This sentence might be understood as being a bit contradictory with

the new text added just above it. It's not entirely clear from this sentence if Oja's rule facilitates learning in the last layer because the last layer follows Oja's rule or because the initial layers follow Oja's rule. In other words, is Oja's rule important when acting in the last layer or it facilitates the learning in the last layer because it changed the activity (or representation) in previous (initial) layers? From the results (and the new added text) it is clear that it is the latter (Oja's rule is important in initial layers, not the last one). The main problem here might be my interpretation of the word "embeddings" and the term "learning in the last layer" in this sentence, so I would like to suggest rephrasing it, being more specific (with details).

We agree that the sentence in question was confusing. The sentence was meant to summarize the effects of Oja's rule all together (not just its inclusion in the last layer, which is the topic of that paragraph). And it was meant to convey that Oja's rule *applied in early layers* improves learning performance of the last layer. We moved the sentence to a separate paragraph and rephrased it to be more specific:

In summary, rather than improving alignment, \mathcal{F}^{Oja} applied to hidden layers provides embeddings that facilitate more effective learning.

6. Page 30, Equation below line 700: There are two "right" parentheses ")" misplaced in the first line of the equation, not the second line. Latex code: $(\sum_k (B_{l+1,l})_{i,k})(e_{l+1})_k$ should be $(\sum_k (B_{l+1,l})_{i,k}(e_{l+1})_k)$. Same for the second term. This way all terms with index k are inside the parentheses where the sum over k is placed.

Thank you for catching this error, which we have fixed.

Reviewer 3

The reviewers have addressed my concerns.

1. **The one minor request I have is that I think it is appropriate to report medians (not means) plus CIs since you are not making the assumption of normality. I recommend changing to medians (or consulting a statistician to make sure it is ok if you think it is appropriate to stick with means).**

We consulted with a Statistics colleague in our department. After going through our data with us, the colleague concluded that the mean (instead of the median) is appropriate for presenting our data. We also checked to make sure that plotting the medians in place of the means does not substantially affect our results. In doing so, we found that plots of the medians were very similar to plots of the means.